biomedical engineering/computer modelling and simulation/astrobiology

parameter estimation, finite-element method, surrogate model, machine learning method, inverse problem

**Authors for correspondence:**
Lei Ren
e-mail: renlei1@mail.nwpu.edu.cn
Hao Gao
e-mail: Hao.Gao@glasgow.ac.uk

# Surrogate models based on machine learning methods for parameter estimation of left ventricular myocardium

Li Cai[1,2,3], Lei Ren[1,2,3], Yongheng Wang[1,2,3], Wenxian Xie[1,2,3], Guangyu Zhu[4] and Hao Gao[5]

[1]Xi'an Key Laboratory of Scientific Computation and Applied Statistics, [2]NPU-UoG International Cooperative Lab for Computation and Application in Cardiology , and [3]School of Mathematics and Statistics, Northwestern Polytechnical University, Xi'an 710129, China
[4]School of Energy and Power Engineering, Xi'an Jiaotong University, Xi'an 710049, China
[5]School of Mathematics and Statistics, University of Glasgow, Glasgow G12 8QQ, UK

LR, 0000-0002-7923-7849; HG, 0000-0001-6852-9435

A long-standing problem at the frontier of biomechanical studies is to develop fast methods capable of estimating material properties from clinical data. In this paper, we have studied three surrogate models based on machine learning (ML) methods for fast parameter estimation of left ventricular (LV) myocardium. We use three ML methods named K-nearest neighbour (KNN), XGBoost and multi-layer perceptron (MLP) to emulate the relationships between pressure and volume strains during the diastolic filling. Firstly, to train the surrogate models, a forward finite-element simulator of LV diastolic filling is used. Then the training data are projected in a low-dimensional parametrized space. Next, three ML models are trained to learn the relationships of pressure–volume and pressure–strain. Finally, an inverse parameter estimation problem is formulated by using those trained surrogate models. Our results show that the three ML models can learn the relationships of pressure–volume and pressure–strain very well, and the parameter inference using the surrogate models can be carried out in minutes. Estimated parameters from both the XGBoost and MLP models have much less uncertainties compared with the KNN model. Our results further suggest that the XGBoost model is better for predicting the LV diastolic dynamics and estimating passive parameters than other two surrogate models. Further studies are warranted to investigate how XGBoost can be used for emulating cardiac pump function in a multi-physics and multi-scale framework.

# 1. Introduction

Heart disease, such as myocardial infarction, has been seriously affecting the quality of life of human beings [1]. Early diagnosis and treatment can effectively reduce the incidence and the mortality of heart disease [2]. In recent years, with the development of medical image technology, such as magnetic resonance imaging (MRI), it has provided paramount data for describing the structure and function of human heart non-invasively [3]. While imaging data alone does not tell the whole story of heart function, e.g. the stress and myocardial stiffness, etc., mathematical modelling and numerical simulation of cardiac function, broadly termed as *in silico* medicine, have been considered as the next generation of medicine for deciphering the mechanism of heart function physiologically and pathologically [4,5].

Finite-element method (FEM) has been successfully applied to cardiac modelling in the past several decades [6–10]. Myocardial material is generally considered to be a hyper-elastic material with a strong nonlinear anisotropic stress response [11]. Many constitutive laws have been used to describe myocardial material behaviours, including isotropic models, transversely isotropic models and, more recently, orthotropic models [12]. In particular, the Holzapfel–Ogden law is a structure-based orthotropic constitutive law that not only accurately describes the mechanical behaviour of the myocardium well from various experimental data [13], but also has been successfully applied to subject-specific cardiac models purely based on *in vivo* routine imaging data [14].

In general, material parameter estimation of a FEM heart model is formulated as an inverse problem [14–18]; for example, it requires solving a constrained optimization problem [19,20] by minimizing the mismatch between limited measured data and the FEM model predictions through finding potential material parameters. This constrained optimization problem can be solved by some gradient-based methods [21], such as the Newton method, the conjugate gradient method, and intelligent methods (also known as nature-inspired methods) [22,23], e.g. the genetic algorithms [24] and the particle swarm method [25]. The gradient-based optimization methods are generally easy to implement and may converge quickly, but cannot guarantee a global optimization. On the other hand, although the evolutionary methods converge slowly, especially for higher-dimensional problems, they can theoretically guarantee a global optimization. When the actual forward problem (high-fidelity model) is computationally intensive and difficult to solve, it further leads to significant computational demand for inferring unknown parameters. To reduce the computational expenses in a constrained optimization problem, surrogate models, which are statistical approximations of the forward problems have been developed for fast parameter inference in biomechanical cardiac models (examples can be found in [26,27]). By using surrogate models, rapid solutions can be quickly computed instead of simulating the computationally expensive FEM model which will speed up the optimization process dramatically, potentially in real time. Surrogate models have also been widely used in certain engineering problems, like aerospace systems among other disciplines [28,29].

Recent studies have shown that myocardial property could be a potential biomarker of predicting ventricular pump function recovery post-myocardial infarction [14]. Estimation of myocardial material parameters from image-based models has attracted intensive interest by formulating a gradient-based inverse problem [15] or using machine learning (ML)-based surrogate approaches for fast parameter inferences [11,27]. By using ML models, the behaviours of the left ventricular (LV) in response to changes in material properties, loads and boundary conditions etc. can be predicted in real time, and can be further applied to the design of medical instruments and monitoring heart condition [30]. For instance, Liang *et al.* [31] have been the first put forward the deep learning technique, a multilayer neural network, as a surrogate of FEM for stress analysis, and the trained model was capable of predicting the stress distributions of aorta by replacing the complex structural finite-element analysis with an average error of less than 1% for ML-predicted stress distribution. Dabiri *et al.* [30] adopted eXtreme Gradient Boosting (XGBoost) and Cubist to predict the LV pressures, volumes and stresses by training them with hundreds of forward FEM simulations of a biomechanical LV model, and their results showed that the surrogate ML models can predict LV mechanics very accurately and are much faster than the FEM models. However model calibration using the ML-based surrogate model has not been carried out in these two studies [30,31]. Di Achille *et al.* [32] inferred the unloading LV geometry used Gaussian process and further statistically learned the infarct shape and size on LV performance in patients extracted from a public database [33]. More recently, Longobardi *et al.* [34] predicted left ventricular contractile function via Gaussian process emulation in aortic-banded rats, the Bayesian history matching was applied to constrain the initial parameter sets in order to exclude

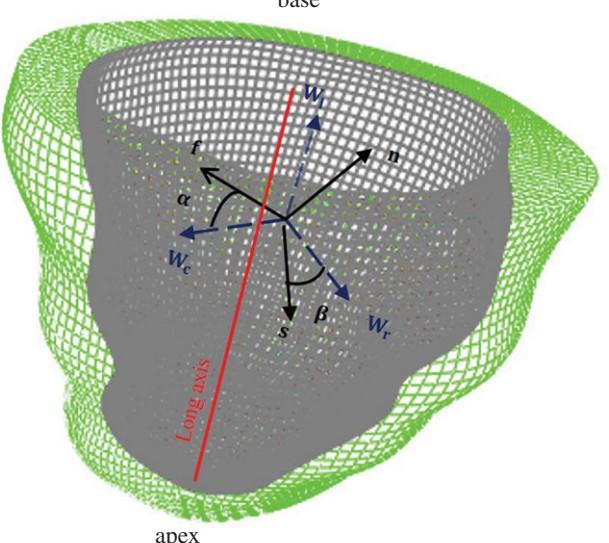

**Figure 1.** Visualization of the LV geometry. (**f**, **s**, **n**) are the fibre, sheet and sheet-normal axes, as described in the text, and (**W**$_c$, **W**$_l$, **W**$_r$) are coordinate axes that indicate the local circumferential, longitudinal and radial axes. The helix angle $\alpha$ is defined to be the angle between **f** and **W**$_c$ in the plane spanned by **W**$_c$ and **W**$_l$, and the sheet angle $\beta$ is defined to be the angle between **s** and **W**$_r$ in the plane spanned by **W**$_l$ and **W**$_r$. The grey colour represents the epicardium and the green colour represents the endocardium.

those points which generate non-physiological biomechanical models. They further performed a Sobol sensitivity analysis using the trained emulator. Most of the studies mentioned above have been focused on demonstrating the feasibility, reliability and accuracy of emulating the cardiac models with various ML approaches, and only a few studies have investigated how the inference problem of unknown parameters can be accelerated using ML surrogate models. For example, Noe *et al.* [26] and Davies *et al.* [27] both have presented a statistical emulation framework for emulating LV mechanics using Gaussian process aiming for accelerating the parameter estimation of myocardium from *in vivo* data. Tens of thousands of simulations of a LV biomechanical model have been performed to train their statistical emulation framework. In both studies, LV cavity volume and 24 circumferential strains at end-diastole were used for training the emulator, but not the dynamics in diastole. They also have demonstrated that the computational costs can be reduced by about three orders of magnitude.

In this paper, we develop three ML-based surrogate models for LV passive mechanics in diastole for model calibration. Firstly, the original high-dimensional material parameter space is projected to a low-dimensional space following the previous studies [27], and the training data are obtained by running forward simulations with sampled parameters from this low-dimensional space. Then ML methods are tuned to learn the relationships between pressure–volume and pressure–strain, respectively. This is different from the studies of [26,27] in which only the end-diastolic state was learned. Finally, the trained surrogate models are applied to formulate an inverse problem for estimating passive parameters of myocardium.

# 2. Methods

## 2.1. Biomechanical model of LV passive dynamics

In this section, we will introduce LV passive dynamics in diastole. A human LV model from our previous studies are used here as shown in figure 1 with 53 548 nodes and 48 050 hexagonal elements. A rule-based approach is used to generate the layered myofibre structures within the myocardium, they are the fibre direction (**f**), the sheet direction (**s**) and the sheet-normal (**n**). In this work, the fibre angle $\alpha$ linearly rotates from $-60°$ to $60°$ from endocardium to epicardium, and the sheet angle $\beta$ linearly rotates from $-45°$ to $45°$ in a similar way, and **n** = **f** × **s**. Details of the LV model reconstruction can be found in [7,8].

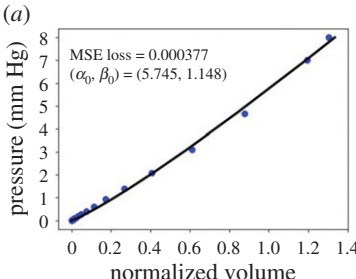
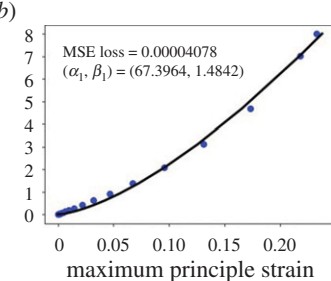
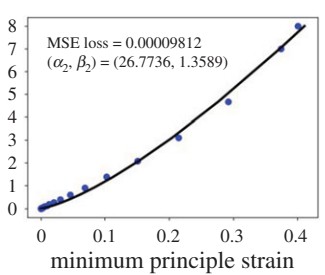

**Figure 2.** A simple example of end-diastolic pressure–normalized volume and pressure–mean principal strain relationships. The discrete points are the original data from the forward simulation, the solid line is the curve fitting to the original data using the expressions in equation (2.4). (a) represents the curve fitting to the pressure and the normalized volume. (b) and (c) represent the curve fitting between the pressure and the mean maximum, minimum principal strains, respectively.

The myocardium is described by a nearly incompressible orthotropic hyper-elastic material strain energy function ($\Psi$) developed in [12], namely the H-O law,

$$\Psi = \frac{a}{2b}\{\exp[b(I_1 - 3)] - 1\} + \sum_{i=f,s}\frac{a_i}{2b_i}\{\exp[b_i(I_{4i} - 1)^2] - 1\}$$
$$+ \frac{a_{fs}}{2b_{fs}}[\exp(b_{fs}I_{8fs}^2) - 1] + \frac{1}{2}K(J - 1)^2, \tag{2.1}$$

where a, b, $a_f$, $b_f$, $a_s$, $b_s$, $a_{fs}$, $b_{fs}$ are material parameters, the term $(1/2)K(J-1)^2$ accounts for the incompressibility of myocardium, and $K$ is a constant bulk modulus ($10^6$ Pa). $I_1$, $I_{4i}$, $I_{8fs}$ ($i = f, s$) are the invariants along myofibre, sheet and sheet-normal directions, respectively,

$$\left.\begin{aligned} I_1 &= \text{tr}(\mathbf{C}), \\ I_{4f} &= \mathbf{f_0} \cdot (\mathbf{Cf_0}), \\ I_{4s} &= \mathbf{s_0} \cdot (\mathbf{Cs_0}) \\ I_{8fs} &= \mathbf{f_0} \cdot (\mathbf{Cs_0}), \end{aligned}\right\} \tag{2.2}$$

and

in which $\mathbf{C} = \mathbf{F}^T\mathbf{F}$ is the right Cauchy–Green deformation tensor and $\mathbf{F}$ is the deformation gradient. $\mathbf{f_0}$, $\mathbf{s_0}$ and $\mathbf{n_0}$ are the layered fibre structure in the reference configuration. In the current configuration, the fibre structure is defined as

$$\mathbf{f} = \mathbf{F}\mathbf{f_0}, \quad \mathbf{s} = \mathbf{F}\mathbf{s_0}, \quad \mathbf{n} = \mathbf{F}\mathbf{n_0}. \tag{2.3}$$

The passive response of the LV dynamics in diastole is implemented and solved using the finite-element (FE) method in a general-purpose FE package ABAQUS (Simulia, Providence, RI, USA). The LV basal surface is fixed in the long-axial direction ($\mathbf{W}_l$-axis) and the circumferential direction ($\mathbf{W}_c$-axis), but allowing radial expansion, see figure 1. A linearly ramped pressure from 0 to 8 mm Hg is applied to the endocardial surface with 25 equal loading steps, and results are saved at each step. The LV cavity volume and principal strains at certain locations are chosen from the forward FE simulations, they are the maximum principal strain ($\varepsilon_{max}$), which is related to myofibre stretch, and the minimum principal strain ($\varepsilon_{min}$), which is related to wall thinning in diastole. In detail, to extract principal strains, 20 locations within the LV wall are randomly chosen using *random* function in Matlab [35], and then the maximum and minimum principal strains are spatially averaged at each loading step. Note that we only select 20 random positions once, and the same 20 positions are used for different simulations to extract strain data. The ventricular cavity volume is the volume enclosed by the endocardial surface. The scatter point in figure 2 shows the relationships between the pressure and the cavity volume, the mean maximum and minimum principal strains from one simulation in diastole. Published studies have found that exponential functions can characterize the nonlinear relationship between the pressure and the LV cavity volume very well [36]. For example, based on *ex vivo* human heart experiments, Klotz *et al.* [36] found that the relationship between the normalized volume ($v_n$) and the loaded pressure ($p$) can be approximated with $p = A_n v_n^{B_n}$, in which $A_n$ and $B_n$ are coefficients, and both are almost invariant among subjects and species. Thus, in this study, we assume the relationships between the pressure and the LV cavity volume, the mean maximum and minimum

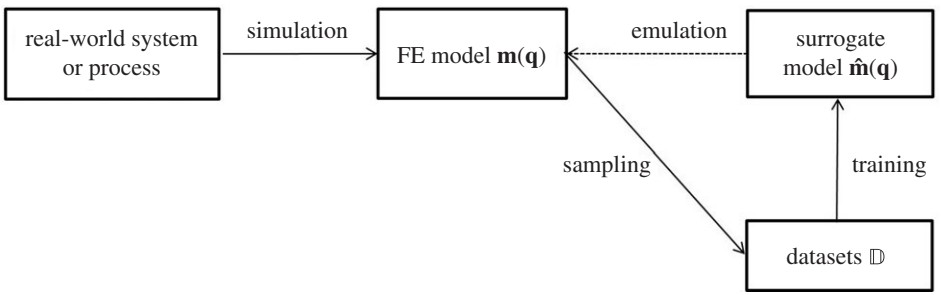

**Figure 3.** The main idea of surrogate model. First, the real-world system, that is the diastolic filling of the left ventricle, can be simulated using the forward FE model $\mathbf{m}(\mathbf{q})$. Then a sampling process is used to collect the datasets $\mathbb{D}$ for training the surrogate model $\hat{\mathbf{m}}(\mathbf{q})$. Finally, the well-trained surrogate model is used to emulate the FE model for approximating the real-world system or process.

principal strains also comply with the exponential function, as suggested by Klotz *et al.* [36], specifically

$$ p = \alpha_0 \, v_\mathrm{n}{}^{\beta_0}, \quad p = \alpha_1 \, (\bar{\varepsilon}_\mathrm{max})^{\beta_1}, \quad p = \alpha_2 \, |\bar{\varepsilon}_\mathrm{min}|^{\beta_2}, \tag{2.4} $$

in which $v_\mathrm{n} = (v - v_0)/v_0$ is the normalized volume with respect to the initial value $v_0$, $\bar{\varepsilon}_\mathrm{max}$ and $\bar{\varepsilon}_\mathrm{min}$ are the mean maximum and minimum principal strains at chosen 20 positions. $\alpha_0$ and $\beta_0$ can be least-square fitted to the $p$–$v_\mathrm{n}$ curve. $\alpha_1$ and $\beta_1$ are derived from the $p$–$\bar{\varepsilon}_\mathrm{max}$ curve, and $\alpha_2$ and $\beta_2$ are derived from the $p$–$\bar{\varepsilon}_\mathrm{min}$ curve. Because the minimum principal strain is negative, we take its absolute value in equation (2.4). Figure 2 shows the results from one simulation, the $p$–$v_n$, $p$–$\bar{\varepsilon}_\mathrm{max}$ and $p$–$\bar{\varepsilon}_\mathrm{min}$ are all fitted well with equation (2.4). Therefore, the output features from a forward ABAQUS simulation are reduced to three pairs of data for describing LV dynamics in diastole, rather than three different curves discretized with 75 data points.

It has been shown that there is a strong correlation among the eight parameters in equation (2.1) [35], thus, it can be very challenging to uniquely determine all eight parameters by using only end-diastolic strains and volume. Following two recent studies from Noe *et al.* [26] and Davies *et al.* [27], the eight-dimensional parameter space is projected into a four-dimensional space,

$$ \left.\begin{array}{ll} a = q_1 \, a_0, & b = q_1 \, b_0, \\ a_\mathrm{f} = q_2 \, a_{\mathrm{f}0}, & a_\mathrm{s} = q_2 \, a_{\mathrm{s}0}, \\ b_\mathrm{f} = q_3 \, b_{\mathrm{f}0}, & b_\mathrm{s} = q_3 \, b_{\mathrm{s}0}, \\ a_\mathrm{fs} = q_4 \, a_{\mathrm{fs}0}, & b_\mathrm{fs} = q_4 \, b_{\mathrm{fs}0}, \end{array}\right\} \tag{2.5} $$

where $\mathbf{q} = (q_1, q_2, q_3, q_4) \in [0.1, \ 5]^4$ are the reduced parameters, $a_0 = 0.22$ kPa, $b_0 = 1.62$, $a_{\mathrm{f}0} = 2.43$ kPa, $b_{\mathrm{f}0} = 1.83$, $a_{\mathrm{s}0} = 0.39$ kPa, $b_{\mathrm{s}0} = 0.77$, $a_{\mathrm{fs}0} = 0.39$ kPa, $b_{\mathrm{fs}0} = 1.70$ are the empirical reference values for a healthy LV model [37]. The range of $\mathbf{q}$ is adopted from [26,27] which was derived from the population average values reported in [37].

## 2.2. Surrogate model

One of the major obstacles of finite-element modelling in cardiac mechanics is the long computational time and tremendous computational resources, in particular if an inverse problem is formulated using finite-element cardiac models. One way of alleviating this burden is to use ML surrogate models [28,38,39]. The basic idea of surrogate model is to construct a measurable functional $\hat{\mathbf{m}}(\mathbf{q})$ to statistically approximate the underlying dynamical model $\mathbf{m}(\mathbf{q})$, the FE LV model in this study, based on costly sampling to construct the dataset $\mathbb{D}$. Whenever a new forward simulation is needed later, the costly function evaluation of $\mathbf{m}(\mathbf{q})$ can be replaced by a fast prediction from the surrogate model $\hat{\mathbf{m}}(\mathbf{q})$. Figure 3 shows the concept of surrogate models in relation to the forward simulator.

### 2.2.1. Design of sampling

In order to fit the surrogate model to the simulator, the ABAQUS FE LV model, we first need to generate the dataset $\mathbb{D}$. For each forward simulation, we consider the input features are the reduced four parameters defined in equation (2.5), and the output features are three pairs of positive values ($\alpha_0$, $\beta_0$),

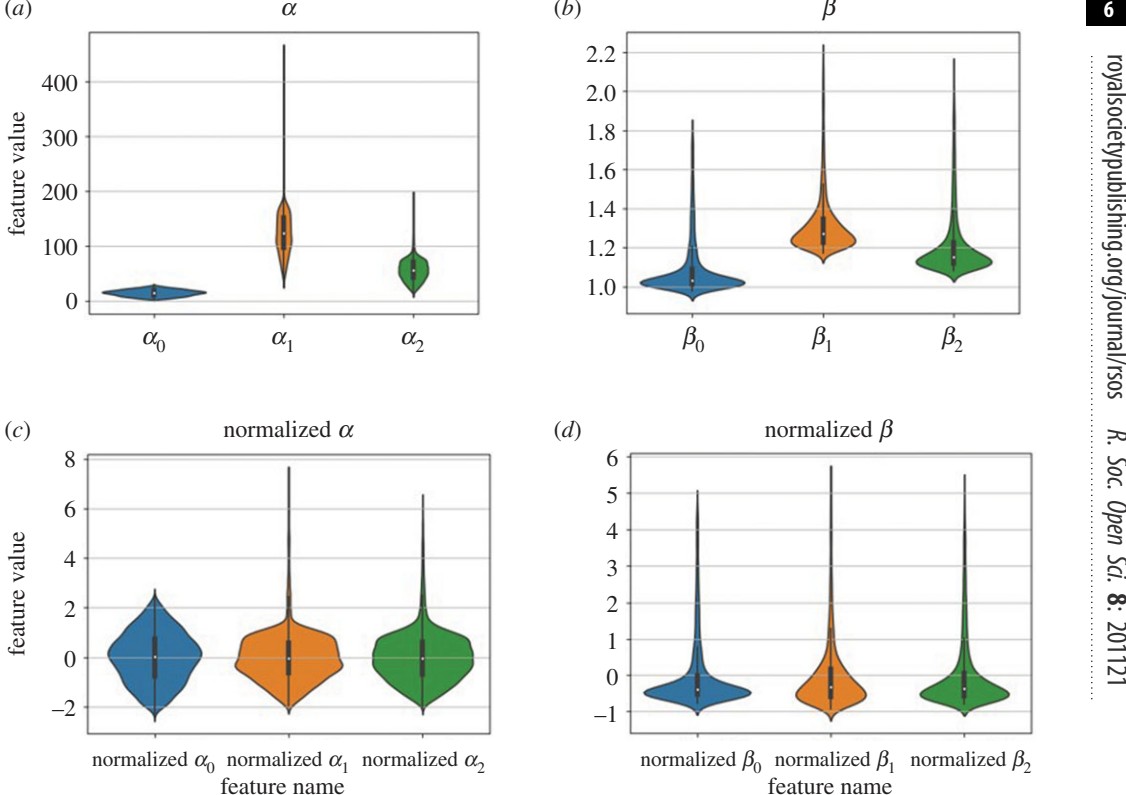

**Figure 4.** Distribution of six outputs of 10 000 samples. (*a*) and (*b*) show violin plots of six output features, (*c*) and (*d*) show the violin plots of six output features after normalization.

($\alpha_1$, $\beta_1$) and ($\alpha_2$, $\beta_2$) as defined in equation (2.4). Then the datasets can be described as

$$\mathbb{D} = \{(\mathbf{q}^i, \ \mathbf{y}^i) | \ \mathbf{q}^i \in [0.1, 5]^4, \mathbf{y}^i \in (0, \ +\infty)^6, \ i = 1, \ 2, \ \dots, n\}, \tag{2.6}$$

where $n$ is the total number of samples. The input features $q_j^i$ represents the j-th feature from i-th sample, and the output features are ordered as $\mathbf{y}^i = (y_1^i, \ \dots, y_6^i) = (\alpha_0^i, \ \beta_0^i, \ \alpha_1^i, \ \beta_1^i, \ \alpha_2^i, \ \beta_2^i)$.

Commonly used sampling methods are the grid sampling, the uniform sampling, and the Latin hypercube sampling, etc. [28]. The most straightforward way of sampling is using a rectangular grid of points or from a uniform distribution. However, this can easily lead to points being clustered together, and causing an ineffective coverage of the parameter space. The Latin hypercube sampling [38] is a statistical method for generating a near-random sample of parameter values from a multidimensional distribution. In brief, when sampling a function of $N$ variables, the range of each variable is divided into $M$ equally probable intervals, $M$ sample points are then placed to satisfy the Latin hypercube requirements. This will ensure from each placed hypercube we could travel the function space along any direction parallel with any of the axes without encountering any other placed hypercube. This is one of the main advantages of this sampling scheme. In this study, we generate 10 000 samples using the Latin hypercube sampling, of which 90% are used for training and the remainder are for testing.

### 2.2.2. Strategy for learning output features

Figure 4*a,b* shows the distributions of each feature from all 10 000 samples with violin plots. Because the FE LV model is nonlinear, the output features are skewed except for $\alpha_0$. Overall, mean square errors (MSEs) of fitting to the volume, the mean maximum and minimal principal strains are less than $10^{-4}$, indicating LV diastolic dynamics from the ABAQUS simulator can be well characterized by the three pairs of features $\alpha_0$, $\beta_0$, $\alpha_1$, $\beta_1$, $\alpha_2$ and $\beta_2$. This is a massive dimension reduction for describing LV dynamics in diastole. We find that the range of these six output features can be very different, specifically the ranges of $\alpha_0$, $\alpha_1$ and $\alpha_2$ are in hundreds and the ranges of $\beta_0$, $\beta_1$ and $\beta_2$ are from 1 to 3. To balance the output features, the general methods include feature scaling and normalization [40]. We then normalize the output features by expressing them as $(y - \bar{y})/(\text{s.d.}(y))$, in

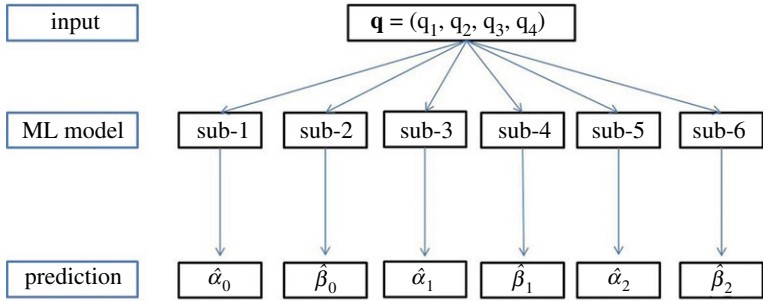

**Figure 5.** The structure of the multiple ML framework. Each ML model contains six sub-models, and each sub-model predicts exactly one output feature.

which $\bar{y}$ and s.d.$(y)$ represent the mean and standard deviation of $y$, the normalized features are shown in figure 4c,d. Finally a ML model is trained for each output feature. Figure 5 illustrates this multiple ML framework; in total six sub-models are trained separately. Note that each sub-model can have different hyper-parameters.

In this study, we will use three ML models, namely the K-nearest neighbour (KNN) [41], XGBoost [30,42], and multilayer perceptron (MLP) [43]. The three ML models can be considered to be supervised learning regression problem [44]. The reasons of choosing these three ML methods are (i) the samples in this work are limited with a few input and output features; (ii) the functional form shown in figure 2 is relatively simple, which may suggest that simple ML models can be applicable; and finally (iii) these selected ML models are easy to implement and readily available in various open-source packages, but they have not been applied in LV model calibration.

### 2.2.3. K-nearest neighbour

Nearest neighbour methods are based on a simple idea by treating the training set as the model, and making prediction of new points based on how close they are to those in the training set. One natural way is to make prediction using the closest training data points, while most datasets contain some degree of noise, a more common method would be to take a weighted average of a set of K nearest neighbours [41,45]. KNN method is a basic classification and regression method, the choice of K value, distance metric and decision rule are three basic hyper-parameters [46]. For the training data $\mathbf{q}^1$, $\mathbf{q}^2$, ..., $\mathbf{q}^n$, and the corresponding target values $y^1, y^2, ..., y^n$ ($y^i$ is denoted as any feature in $\mathbf{y}^i$), the prediction in a new point $\mathbf{q}^p$ can be realized by firstly searching a set of K nearest neighbours in the training set, and then $\hat{y}^p$ is given by a weighted averaged of those K nearest neighbours, that is

$$\hat{y}^p = \sum_{i=1}^{K} \omega_i y^i, \tag{2.7}$$

in which $\omega_i$ is the weight for the i-th nearest neighbour. Note the closer the point to the predicted point, the greater weight this point takes, and $\omega_i$ is defined as

$$\omega_i = \frac{1/L(\mathbf{q}^i, \mathbf{q}^j)}{\sum_{j=1}^{K} 1/L(\mathbf{q}^j, \mathbf{q}^p)}, \tag{2.8}$$

in which $L$ is a distance function, which can be either as $L_1$ or $L_2$

$$L_1(\mathbf{q}^i, \mathbf{q}^j) = \sum_{k=1}^{4} |q_k^i - q_k^j| \quad \text{or} \quad L_2(\mathbf{q}^i, \mathbf{q}^j) = \sqrt{\sum_{k=1}^{4} (q_k^i - q_k^j)^2}. \tag{2.9}$$

### 2.2.4. eXtreme Gradient Boosting

Tree boosting is a highly effective and widely used machine learning method, in particular XGBoost (eXtreme Gradient Boosting), a boosting tree model developed by Chen & Guestrin [42], an implementation based on gradient boost decision tree [47], which boosts many tree models to form a strong regression model. In XGBoost, the additive strategy is used to predict output variables [48],

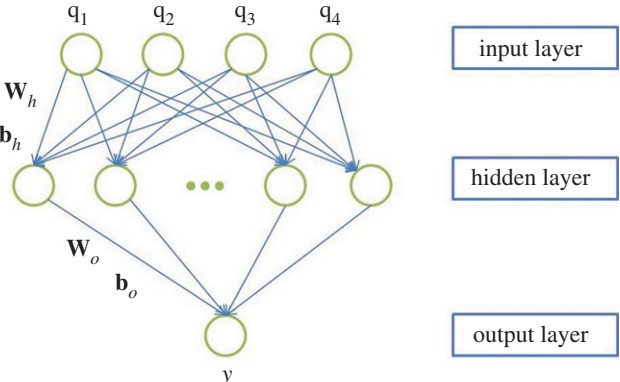

**Figure 6.** A schematic of a multilayer perceptron model with one single hidden layer.

learning one classification and regression tree (CART) [49] per iteration to fit the residual of the predicted results of the previous trees to the true values of the training samples. Note the prediction value in $\mathbf{q}^p$ at step t as $\hat{y}^{p(t)}$. Then we can write each sub-model in the form

$$\hat{y}^{p(t)} = \sum_{j=1}^{t} g_j(\mathbf{q}^p), \; g_j \in \mathcal{G}, \tag{2.10}$$

where $t$ is the number of CARTs, $g_j$ is a function in the functional space $\mathcal{G}$, the set of all possible regression trees. The loss function to be optimized is given for each feature, which is defined as

$$\mathcal{L}_{XGB}^{(t)} = \sum_{i=1}^{n} l(y^i, \hat{y}^{i(t)}) + \sum_{j=1}^{t} \Omega(g_j) = \sum_{i=1}^{n} l(y^i, \hat{y}^{i(t-1)} + g_t(\mathbf{q}^i)) + \Omega(g_t) + \text{constant}, \tag{2.11}$$

where $l$ is a distance function that measures the distance between the prediction $y^{i(t)}$ and the target $\hat{y}^i$. The term $\Omega$ penalizes the complexity of the model. We take the Taylor expansion of the loss function up to the second order, that is

$$\mathcal{L}_{XGB}^{(t)} = \sum_{i=1}^{n} \left[ l(y^i, \hat{y}^{i(t-1)}) + p_i g_t(\mathbf{q}^i) + \frac{1}{2} h_i g_t^2(\mathbf{q}^i) \right] + \Omega(g_t) + \text{constant}, \tag{2.12}$$

where $p_i$ and $h_i$ are the first and second order derivatives of $l(y^i, \hat{y}^{i(t-1)})$. After we remove all the constants, the specific objective at step $t$ becomes

$$\mathcal{L}_{XGB}^{(t)} = \sum_{i=1}^{n} \left[ p_i g_t(\mathbf{q}^i) + \frac{1}{2} h_i g_t^2(\mathbf{q}^i) \right] + \Omega(g_t). \tag{2.13}$$

For more details, please refer to references [42,47].

### 2.2.5. Multilayer perceptron

MLP is a class of feed-forward artificial neural network [43], consisting of at least three layers of nodes: an input layer, a hidden layer and an output layer. Fully connected multiple hidden layers can be added to the hidden layer. Except for input nodes, each node in MLP is a neuron with a nonlinear activation function. A feed-forward neural network should have a linear output layer, and at least one hidden layer of an activation function. Universal approximation theorem [50] indicates that if we give the network a sufficient number of hidden layers, it can approximate the Borel measurable function of any finite dimensional space to another finite dimensional space. In this sense, multilayer feed-forward networks are a class of universal approximators. In this work, we adopt a MLP model with only one hidden layer [51] shown in figure 6. The input layer has four nodes which will take $q_1$, $q_2$, $q_3$, $q_4$ in the reduced space, and the output layer is combined to output one select feature.

The rectified linear unit (ReLU) is used for the activation function at each node of the hidden layer [52], which can be described as

$$\sigma(\mathbf{q}) = \max(0, \, \omega_h^T \cdot \mathbf{q} + \mathbf{b}_h), \tag{2.14}$$

in which $\omega_h$ and $\mathbf{b}_h$ represent weights and bias between the input layer and the output layer. Because the MLP model only contains one hidden layer, the prediction for each feature $y^p$ can be represented as

$$
\begin{aligned}
\hat{y}^p &= \omega_o^T \cdot \sigma(\mathbf{q}^p) + \mathbf{b}_o \\
&= \omega_o^T \cdot \max(\mathbf{0}, \omega_h^T \cdot \mathbf{q}^p + \mathbf{b}_h) + \mathbf{b}_o,
\end{aligned}
\tag{2.15}
$$

where $\omega_o$ and $\mathbf{b}_o$ are the weights and biases between the hidden layer and the output layer. The loss function for the MLP model is defined using the mean square error (MSE) with $L_2$ penalty (regularization term) to prevent over-fitting,

$$
\mathcal{L}_{MLP} = \frac{\sum_{i=1}^{n} (y^i - \hat{y}^i)^2}{n} + \lambda(\|\omega_o\|_2 + \|\omega_h\|_2),
\tag{2.16}
$$

where $\mathbf{y} = \{y^i, \ i = 1, \ldots, n\}$ are the true values, $\hat{\mathbf{y}} = \{\hat{y}^i, \ i = 1, \ldots, n\}$ are the predictions from the MLP model, $\lambda$ is a penalty parameter and we fix it as $1 \times 10^{-4}$, and $\|\cdot\|$ denotes the $L_2$ norm in Euclidean space.

## 2.3. Training machine learning-based surrogate models

The KNN and MLP models are implemented using Scikit-learn library[1] [53]. Specifically, the KNN model uses a KDTree algorithm [54] (a fast algorithm to generalize N-point problems) to compute the nearest neighbours. The MLP model uses an Adam solver [55] (a stochastic gradient-based optimizer) to optimize the MSE with $L_2$ penalty (see equation (2.16)). Early stopping is adopted to terminate training when validation score is not improved after 10 epochs. The maximum number of iterations is set to 200. The XGBoost model is implemented using the XGBoost Python library.[2] The tolerance for the optimization is $1 \times 10^{-4}$.

A competent ML model not only fits the training data well, but also should have a good predictability for the test data, the unseen situations. The choice of hyper-parameters can significantly affect the performance of ML models, while determining most optimal hyper-parameters can be very complex [56]. In this paper, a grid search method with a fivefold cross-validation is adopted [57] for its easy implementation. The coefficient of determination $R^2$ is used to evaluate the predictability of a ML model for choosing the optimal set of hyper-parameters similar to in [30],

$$
R^2(\mathbf{y}, \hat{\mathbf{y}}) = 1 - \frac{\sum_{i=1}^{n} (y^i - \hat{y}^i)^2}{\sum_{i=1}^{n} (y^i - \bar{y})^2}, \quad \text{and} \quad \bar{y} = \frac{1}{n}\sum_{i=1}^{n} y^i.
\tag{2.17}
$$

$R^2$ represents the proportion of variance that has been explained by input parameters in a model. A score of 1.0 indicates a perfect prediction, and it may be negative because a model can be arbitrarily worse, see Scikit-learn documentation [53].

Tuning hyper-parameters using the grid search method in a multidimensional space can still be time-consuming (from several hours to a few days). To determine hyper-parameters efficiently [58], we use a step-wise approach by optimizing one hyper-parameter at each time step while fixing others. Taking the MLP model as an example. We first fix initial learning rate (*learning_rate_init*) to be 0.01, then search the number of neurons (*hidden_layer_sizes*) in a predefined grid range to determine its optimal value. We now fix *hidden_layer_sizes* using the best value from the previous step and search the optimal value for *learning_rate_init*. Grid search for other two ML models are the same as the MLP model. All pre-defined grid values for each hyper-parameter are shown in table 1, and the name of hyper-parameter follows the conventions in sklearn and XGBoost libraries.

## 2.4. Parameter inference

Estimating myocardial property from measured volume and strain is challenging because of sparse data and correlation between different parameters, which has been discussed in [35]. In general, an inverse problem is formulated by solving a constrained optimization problem with thousands of forward simulations which may take days and weeks [35,59]. The objective function is usually formulated as the differences between the model predictions and experimental measurements (e.g. volume and

---

[1]https://scikit-learn.org/stable/.

[2]https://github.com/dmlc/xgboost.

**Table 1.** The grid values of hyper-parameters for three ML models.

| ML methods | hyper-parameters | description | grid search values |
|---|---|---|---|
| KNN | *n_neighbours* | number of neighbours | 1, 2, 4, 6, 8, 10 |
| | *p* | power parameter | 1, 2 |
| XGBoost | *max_depth* | maximum tree depth | 4, 6, 8, 16 |
| | *learning_rate* | boosting learning rate | 0.05, 0.1, 0.075 |
| | *n_estimators* | number of boosting trees | 500, 1000, 2000 |
| MLP | *hidden_layer_sizes* | number of neurons | 256, 512, 1024 |
| | *learning_rate_init* | initial learning rate | $5 \times 10^{-3}$, $1 \times 10^{-3}$, $5 \times 10^{-2}$, $1 \times 10^{-2}$ |

strain for the LV model), and constitutive parameters are iteratively updated until the objective function is minimized [19,35].

We further apply the three ML models for myocardial parameter estimation using a set of synthetic data ($v^*$, $\varepsilon^*$) by simulating the FE LV model with known parameters, this will allow us to compare the suitability of the three ML models for parameter inferences. The surrogate models do not directly predict volume and principal strains but six output features (**y**), thus we define the objective function [35] as

$$f(\mathbf{q}) = \frac{1}{6} \sum_{j=1}^{6} (y_j - y_j^*)^2,$$  (2.18)

where $y_j$ represents j-th output feature, $y_j^*$ are the corresponding output feature obtained from the synthetic data ($v^*$, $\varepsilon^*$) according to equation (2.4). Note that the output features can be readily converted into the actual cavity volume and principal strains as suggested in equation (2.4). A differential evolution algorithm [60] is employed for inferring **q** using ML-based surrogate models. Differential evolution is a stochastic population-based method which has been applied for global optimization problems. The optimization procedure is implemented using the Python library *scipy.optimize*.[3] We choose that the total population size is 50, the relative tolerance for convergence is $1 \times 10^{-6}$, the maximum number of iterations is 500, the mutation constant is in (0.5, 1), the recombination constant is 0.7, and the initial values are randomly selected from the range [0.1, 5][4].

We further infer **q** by formulating an inverse problem using the forward ABAQUS FE LV model as shown in figure 7. Specifically,

  (i) randomly initializing constitutive parameters from the predefined range,
 (ii) running the FE LV model with ramped blood pressure till end-diastole,
(iii) fitting the pressure–volume and pressure–strain according to equation (2.4),
(iv) computing the objective function shown in equation (2.18) to determine whether the algorithm has converged (i.e. exceeding maximum iteration number, or less than the predefined error, etc.),
 (v) if the algorithm converges, then the last updated parameters are considered to be the optimal parameters, otherwise going to step (ii) by updating constitutive parameters.

This inverse problem is implemented using *lsqnonlin* in Matlab, which is based on a trust-region-reflective algorithm [61]. Trust-region-reflective is a gradient-based algorithm with fast convergence but it may not guarantee a global optimum. We set the maximum number of iterations 500, the relative tolerance on the target value $1 \times 10^{-6}$, and the initial values are also randomly selected from the range [0.1, 5][4]. We are aware that there are other gradient-free algorithms like the differential evolution used for the ML models. In a study of estimating myocardial parameters, Nair *et al.* [62] reported it took 25–40 days for the inference procedure in a desktop computer using a genetic algorithm. Because of the high computational costs associated with gradient-free algorithms, we decide not to use those algorithms for the inverse problem using the FE LV model.

[3]https://docs.scipy.org/doc/scipy-1.0.0/reference/tutorial/optimize.html.

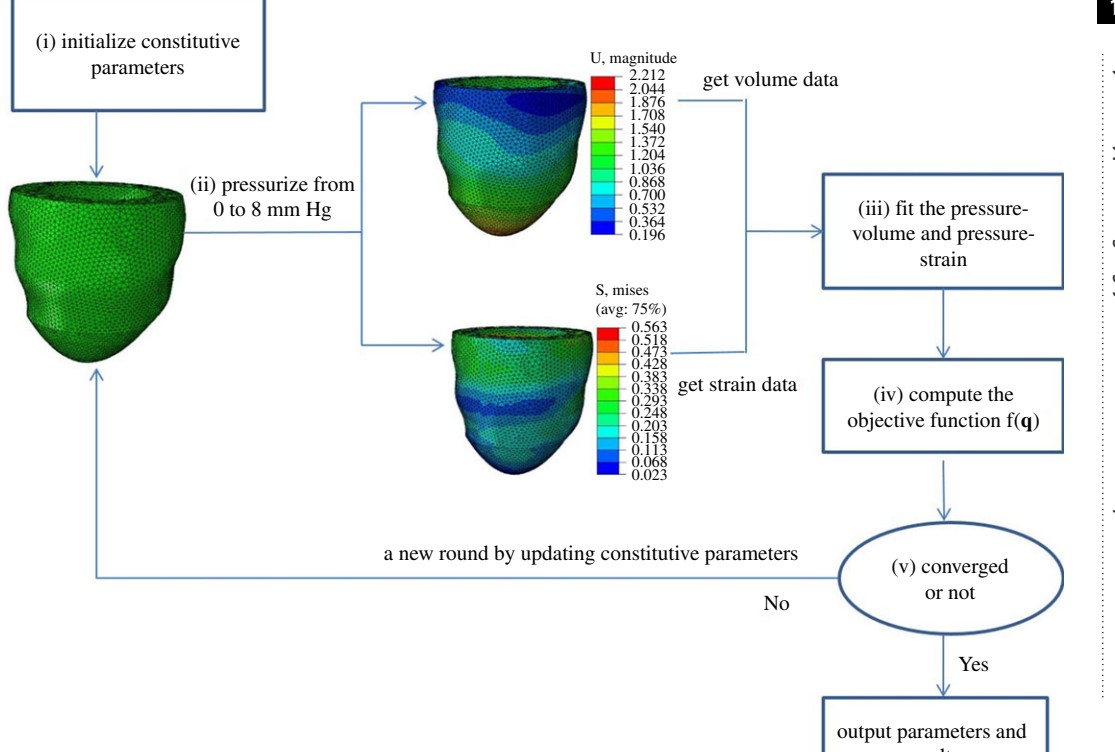

**Figure 7.** The iterative schematic of the inverse problem based on ABAQUS simulation.

**Table 2.** Tuning results of hyper-parameters for six sub-models.

| ML methods | hyper-parameters | best hyper-parameters | | | | | |
|---|---|---|---|---|---|---|---|
| | | $\alpha_0$ | $\beta_0$ | $\alpha_1$ | $\beta_1$ | $\alpha_2$ | $\beta_2$ |
| KNN | n_neighbours | 10 | 8 | 8 | 6 | 8 | 6 |
| | p | 2 | 1 | 1 | 1 | 1 | 1 |
| XGBoost | max_depth | 6 | 6 | 6 | 6 | 4 | 4 |
| | learning_rate | 0.05 | 0.05 | 0.1 | 0.05 | 0.05 | 0.1 |
| | n_estimators | 1000 | 1000 | 1000 | 1000 | 1000 | 1000 |
| MLP | hidden_layer_sizes | 1024 | 1024 | 1024 | 512 | 1024 | 1024 |
| | learning_rate_init | $1 \times 10^{-3}$ | $1 \times 10^{-3}$ | $1 \times 10^{-3}$ | $1 \times 10^{-3}$ | $1 \times 10^{-3}$ | $1 \times 10^{-3}$ |

All the computations are carried out in a 64-bit Windows 7 workstation with 1.7 GHz Intel Core E5-2609 CPU and 32 GB RAM. ABAQUS simulation and training ML models are performed on 16 threadings for parallelism. No parallelization is used when inferring material parameters within *lsqnonlin* and *differential evolution* solvers.

# 3. Results

## 3.1. Machine learning methods and hyper-parameters tuning

Firstly, we adopt fivefold cross-validation to train the ML models on training data based on the MSE loss function. Then, we test every model on test data and select the optimal hyper-parameters from the grid values with the highest $R^2$ score. Note that the higher the $R^2$, the better the model predictability. For each surrogate model, six identical ML sub-models are constructed, and each sub-model has different optimal hyper-parameters as summarized in table 2.

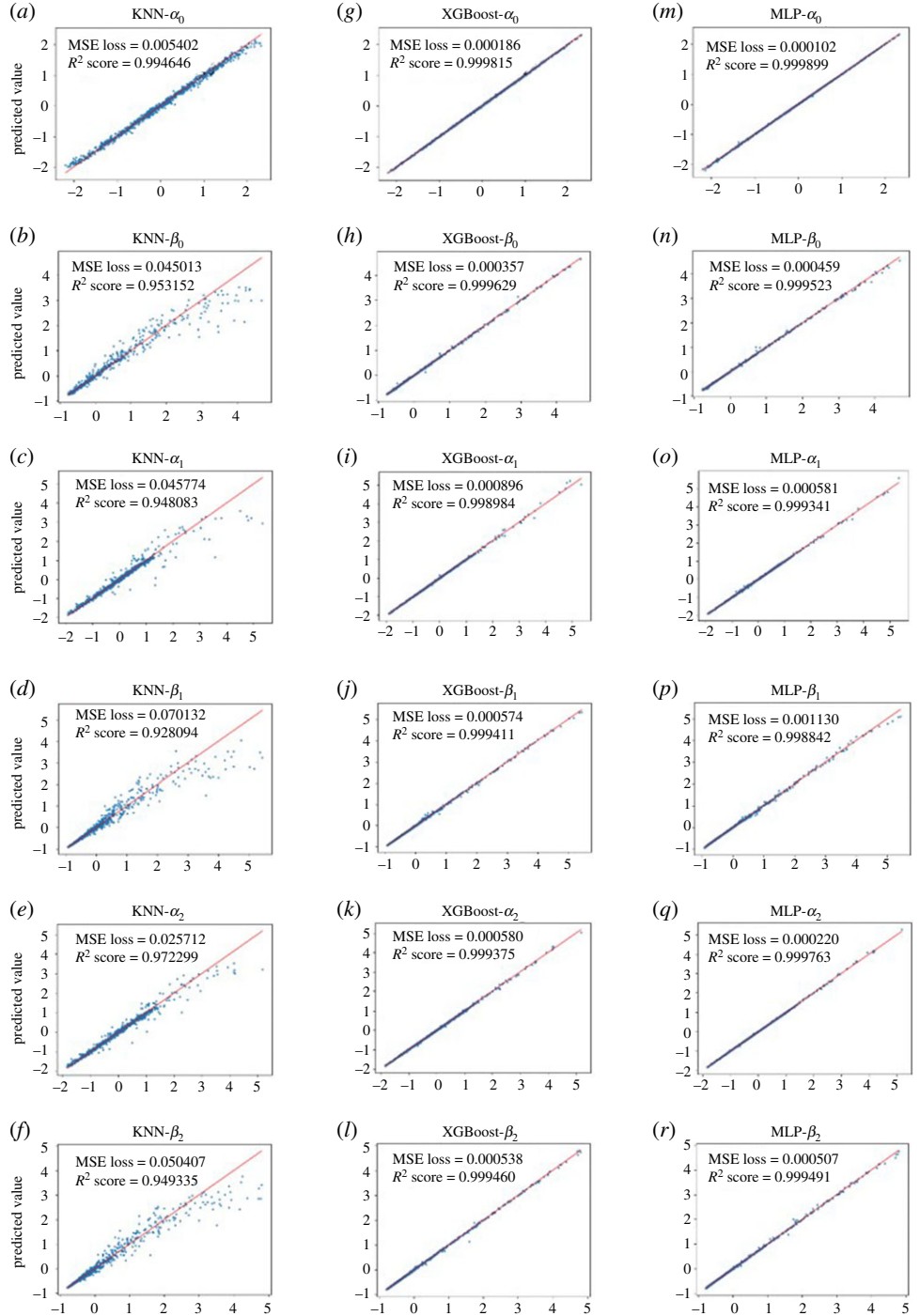

**Figure 8.** Prediction results of six output features from three ML models on test samples. $(a-f)$, $(g-l)$ and $(n-r)$ are the test results of KNN, XGBoost and MLP, respectively. The output features here are normalized as $(y - \bar{y})/(\text{s.d.}(y))$.

Figure 8 shows the predictions of six output features from the three ML models on test samples. The red line suggests the exact predictions of the true values and blue points are the predictions. The closer the blue points to the red line, the better predictability for ML models. It can be found that the XGBoost and MLP models can predict all output features very well in general, very close to the red line. While the predictions from the KNN model are much poorer compared with the XGBoost and MLP models, in particular for large true values. Moreover, the MSEs of the KNN model are nearly 10 orders higher than the other two ML surrogate models, and $R^2$ scores of the XGBoost and MLP models are around 0.999, much higher than the $R^2$ score of the KNN model. Thus the XGBoost and MLP models seem more suitable for learning our FE LV model in diastole compared to the KNN model. For each prediction, the KNN

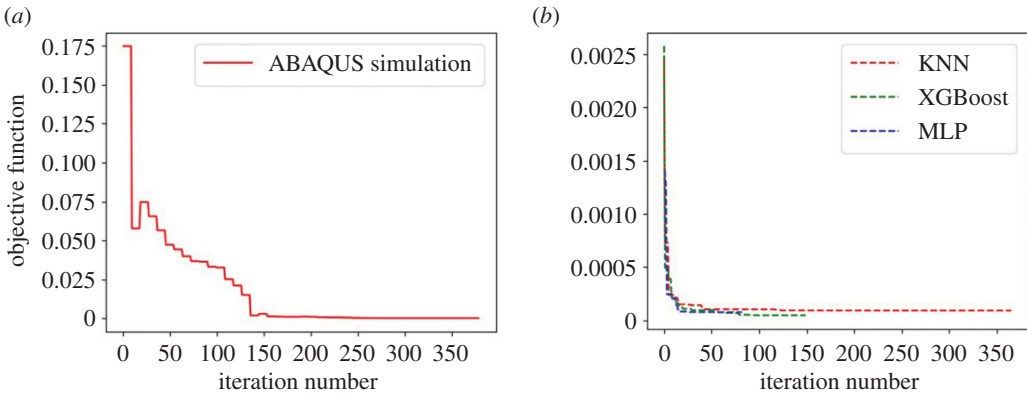

**Figure 9.** The objective function using ABAQUS simulation (*a*) and three ML-based surrogate models (*b*).

**Table 3.** Computation summary of the optimization procedure.

| methods | $f(\mathbf{q})$ | total cost time | time for one run | evaluation numbers |
|---|---|---|---|---|
| ABAQUS | $5.8537 \times 10^{-8}$ | 63 h | 6 min | 378 |
| KNN | $3.1227 \times 10^{-5}$ | 5930 s | 0.08 s | 71 505 |
| XGBoost | $1.6431 \times 10^{-5}$ | 587 s | 0.02 s | 28 805 |
| MLP | $1.7441 \times 10^{-5}$ | 69 s | 0.005 s | 12 805 |

**Table 4.** The estimated parameters using ABAQUS and ML models.

| methods | $q_1$ | $q_2$ | $q_3$ | $q_4$ |
|---|---|---|---|---|
| reference values [35] | 1 | 1 | 1 | 1 |
| ABAQUS | 1.0000 | 0.9999 | 1.0001 | 1.0002 |
| KNN | 0.9634 | 0.9896 | 1.1625 | 1.8253 |
| XGBoost | 1.0228 | 1.0040 | 1.06299 | 0.9689 |
| MLP | 1.0066 | 0.9788 | 1.0869 | 1.2164 |

model takes 0.08 s, and 0.02 s for the XGBoost, and 0.005 s for the MLP model, all are significantly less than the forward ABAQUS simulation, which takes around 6 min for one run. The computational time using the ML models is almost reduced by 3 to 4 orders, and summarized in table 3.

## 3.2. Parameter estimation

To inversely infer unknown parameters using the forward FE LV model as in [35], 378 forward ABAQUS simulations are carried out, which takes about 63 h (almost 3 days). The corresponding objective function with respect to the iterations is shown in figure 9*a*. When using the ML models for inferring unknown parameters, it only take several minutes, especially for the MLP model, which takes only 69 seconds (around 1 min); see table 3. Figure 9*b* shows the objective function with respect to the iterations for the three ML models. The objective function is the smallest when using the XGBoost model, followed by the MLP model and poorest in the KNN model.

Table 4 summarizes the estimated parameters from the four inverse problems. In general, estimated parameters by running ABAQUS simulations are very close to the ground truth values with nearly negligible objective function, but the computational cost is the highest, almost 3 days. The XGBoost seems to have a better performance for inferring parameters compared with the KNN model and the MLP model, with poorest for the KNN model, in particular for $q_3$ and $q_4$. The poorest performance in the KNN model may be partially explained by its poor predictability as shown in figure 8. Compared

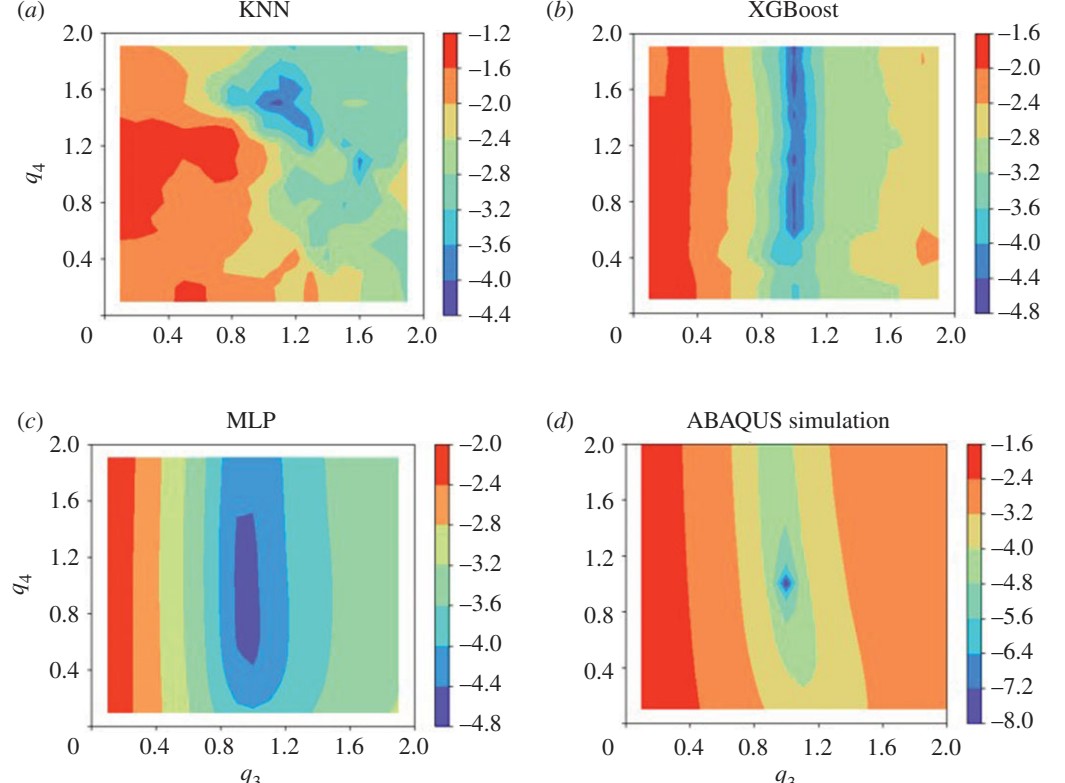

**Figure 10.** Surface plots of logarithmic (to the base 10) objective function by fixing $q_1 = q_2 = 1$ but varying $q_3$, $q_4$ from 0.1 to 2; (a)–(d) represents the surface plots derived from models of KNN, XGBoost, MLP and ABAQUS simulation, respectively.

with $q_1$ and $q_2$, $q_3$ and $q_4$ have a larger discrepancy with respect to the true values when using the three ML models, and also slightly larger when using ABAQUS simulations. This may suggest that the inverse parameter estimation problem by using measurements of LV dynamics in whole diastole may still experience unidentifiable issues as suggested in [15,63], in which only end-diastolic volume and strains were used for inferring unknown parameters. We further plot the objective function by varying $q_3$ and $q_4$ with fixed $q_1 = 1$ and $q_2 = 1$ as shown in figure 10. It can be found that the surface plots of using the XGBoost, MLP and ABAQUS simulation are similar, the minimum from the ABAQUS simulation is clearly identifiable, but a narrow valley parallel to $q_4$ with multiple local minima for the XGBoost, and that valley becomes a flat region for the MLP model. Thus, $q_4$ seems unidentifiable when using the XGBoost and MLP models even with fixed $q_1 = 1$ and $q_2 = 1$. The surface plot of the objective function for the KNN model is very different from the others, which again indicates its poor performance of learning the FE LV model.

Figure 11a,b shows the myofibre stress–stretch relationship under uni-axial stretching mode derived from the H-O law (equation (2.1)) with the parameters in table 4. For the stretch along fibre direction, the inferred stress–stretch data from the three ML models and ABAQUS simulation are very close to the ground truth data. But larger discrepancies can be found for the Cauchy stress along the sheet direction when inferred from the three ML models. Figure 11c,d shows corresponding stress residuals, which are calculated by subtracting the inferred Cauchy stress from the ground truth data. It is not surprising that the inference using ABAQUS simulations achieves the best accuracy because the synthetic data are generated using the ABAQUS simulation. The XGBoost comes the second closest to the ground truth data with smaller residuals compared with the KNN and MLP models.

In order to have an indication of the uncertainty in our inference using the three ML models, we adopt a bootstrap approach [64] for the inverse uncertainty quantification. We first compute the prediction value $\mathbf{y}^{\mathrm{pred}}$ using the estimated parameters in table 4, we then compute the residuals between the referential values $\mathbf{y}^{\mathrm{ref}}$ (i.e. the output features which are obtained from the synthetic LV model), they are $\boldsymbol{\epsilon}$:$\{\epsilon_i = y_i^{\mathrm{ref}} - y_i^{\mathrm{pred}}, (i = 1, 2, 3, 4, 5, 6)\}$. By randomly selecting all combinations from $\boldsymbol{\epsilon}$, and 0 used from unselected residuals as supplement, we can obtain a set of residuals $\mathcal{R} = \{\hat{\boldsymbol{\epsilon}}_J\}_{1 \leq J \leq 64}$, and then generate surrogate data $\hat{\mathbf{y}}_J = \mathbf{y}^{\mathrm{pred}} + \hat{\boldsymbol{\epsilon}}_J$, where $\hat{\boldsymbol{\epsilon}}_J$ is the J-th draw from $\mathcal{R}$. We then repeat the parameter inference on each $\hat{\mathbf{y}}_J$ to obtain new estimations. The mean and standard deviation of all

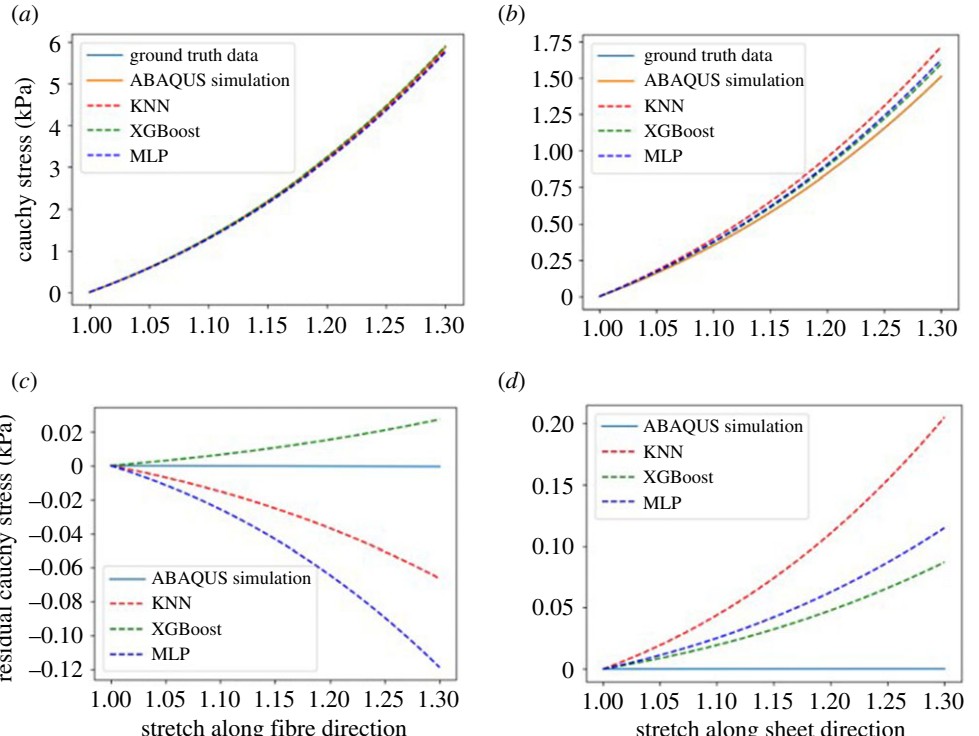

**Figure 11.** Cauchy stress under uni-axial stretch with estimated parameters from ABAQUS simulations and three ML models along fibre (*a*) and sheet (*b*) directions. The corresponding residual Cauchy stress for those four methods along fibre (*c*) and sheet (*d*) directions.

**Table 5.** Mean and standard deviation (s.d.) of estimated parameters and objective function values obtained from the bootstrap approach.

| methods | | $q_1$ | $q_2$ | $q_3$ | $q_4$ | $f(q)$ |
|---|---|---|---|---|---|---|
| KNN | mean | 0.9906 | 1.0292 | 1.1814 | 1.2188 | $7.1862 \times 10^{-5}$ |
| | s.d. | 0.0086 | 0.0356 | 0.5742 | 0.5373 | $5.0849 \times 10^{-5}$ |
| XGBoost | mean | 1.0127 | 0.9668 | 1.0708 | 1.0485 | $2.0316 \times 10^{-5}$ |
| | s.d. | 0.0073 | 0.0194 | 0.1794 | 0.0239 | $1.9800 \times 10^{-5}$ |
| MLP | mean | 0.9934 | 1.0351 | 1.0781 | 1.0959 | $2.9699 \times 10^{-5}$ |
| | s.d. | 0.0091 | 0.0273 | 0.1581 | 0.1701 | $2.7242 \times 10^{-5}$ |

estimated parameters and objective function values are listed in table 5. The uncertainties associating with $q_1$ and $q_2$ are much less than the uncertainties for $q_3$ and $q_4$, again the XGBoost has smaller uncertainties in $q_3$ and $q_4$ compared with the KNN and the MLP model. The large uncertainties associated with the KNN model suggests its unsuitability for inferring constitutive parameters by replacing a FE LV model.

## 4. Discussion

In this study, we have developed three surrogate models based on three ML methods, namely the KNN, XGBoost and MLP. These models are used to estimate the material parameters of LV myocardium using the cavity volume and the maximum and minimum principal strains. The developed ML models are trained through cross-validation, and the average $R^2$ scores for the testing data are very close to one. Comparing with the conventional parameter estimation based on numerical evaluations of the forward model, the ML surrogate models provide a significant improvement in computational time with an acceleration by hundreds or even thousands of times.

Gaussian process also has been applied to parameter estimation in myocardium [26,27] using a synthetic LV model, while their output features were only from the measurements at end-diastolic phase but not for the whole diastole. In a followed study, Lazarus et al. [63] found that because of lacking data, only using end-diastolic measurements makes it challenging to infer $q_3$ and $q_4$. Instead, in this study the parametrized pressure–volume and pressure–strain data in diastole are used for training the ML models and parameter inferences. Still, we observe large uncertainties associating with $q_3$ and $q_4$. Thus extra data will be needed for identifying $q_3$ and $q_4$ with high confidence. We also have trained a Gaussian process model for our data, and the prior covariances of Gaussian process are computed by using a radial-basis kernel function. We then use the trained Gaussian process model for inferring $\mathbf{q}$, the inferred $\mathbf{q}$ is (1.0053, 1.0027, 1.0812, 1.1504) with $\mathbf{f(q)} = 1.7424 \times 10^{-5}$. It is slightly poorer than the XGBoost model as seen in table 4. While this should still suggest that Gaussian process is a good surrogate model as XGBoost for learning LV dynamics in diastole.

In a recent study, Liu et al. [65] has used a two hidden-layer neural network for estimating in vivo constitutive parameters of aortic wall. Considering the LV dynamics in diastole is a passive expansion process with increased volume and strain magnitude as shown in figure 2, a deep neural network may over-fit our data easily because of its large number of degrees of freedom. As a test, we implement a two hidden-layer neural network and train using the same data with a fixing learning rate of $1 \times 10^{-3}$, and the grid search is used for selecting optimal neuron numbers at each hidden layer with the grid values in table 1. Our result shows that the mean $R^2$ score for the testing data is almost the same as using the one hidden-layer neural network, which may suggest that the one-layer MLP can learn the LV FE model in diastole well. For a very complex dynamics system, i.e. whole heart contraction, a deep neural network would be necessary.

There are various methods for hyper-parameter tuning [58], including random search, grid search, Bayesian optimization, etc. Random search uses a randomized search over parameters in which each setting is sampled from a distribution over possible parameter values. Grid search generates candidates from a grid of parameter values. Bayesian optimization [66] is a global optimization method based on sequential training of a statistical approximation (Gaussian process) of the target function. We also apply the Bayesian optimization for hyper-parameter tuning using bayesopt [67] implemented in Python. By comparing with the trained ML models using the grid search approach, the $R^2$ scores are almost the same, while the Bayesian optimization takes a much longer time for tuning ML models than the grid search approach.

Uncertainty quantification (UQ) is an essential step in applying mathematical models for decision making, the LV model included. There are various sources of uncertainty in cardiac models, including intrinsic variability (i.e. collagen content), parameter uncertainty (i.e. myocyte stiffness), initial/boundary condition uncertainty (i.e. ventricular cavity pressure, pericardium), geometry uncertainty (i.e. myocyte architecture), model uncertainty (i.e. continuum approximation of soft tissue), etc. Interested readers refer to [39,68] for recent reviews of uncertainty and variability in cardiac models. If a surrogate model is used for prediction, an additional source of uncertainty is the model discrepancy between the surrogate model and the mathematical model. As evidenced in figure 8, the KNN has the worst prediction due to this model discrepancy, which may potentially explain the large discrepancy and high uncertainty in estimated parameters (table 5). Optimal computational design strategy would be needed for achieve high accuracy for the surrogate model, for example sampling strategies according to the underlying physics, dense coverage for small parameter values. A further model discrepancy is the discrepancy between the forward model and the reality. In a recent study, Lei et al. [69] studied the model discrepancy between the mathematical model and reality in cardiac ionic models. They modelled the discrepancy using Gaussian processes and autoregressive–moving-average model for improved prediction of unseen situations, and found that different methods for accounting model discrepancy are needed for different models in different situations. The similar framework could also be applied to the FE LV model, which is a numerical approximation of LV dynamics under various assumptions (i.e. homogeneous material property, rule-based myofibre structure, ignoring viscous effects).

As shown in table 4, estimated parameters using those three trained surrogate models do not match the ground truth values, and are also inferior to the values inferred directly using the forward ABAQUS LV model. To quantify the associated uncertainty in inferred parameters, we further adopt a bootstrap approach, results are summarized in table 5. In general $q_1$ and $q_2$ can be inferred with high accuracy, but not $q_3$ and $q_4$, especially for the KNN. XGBoost achieves the best performance with very little uncertainties in $q_1$, $q_2$ and $q_4$, but poor in $q_3$; that is partially because $q_3$ is related to nonlinear response of collagen network and myofibres [12], which are not fully engaged under normal

**Table 6.** The MSE values of sample test cases using the three trained ML models.

| methods | MSE loss function | | | | | |
| --- | --- | --- | --- | --- | --- | --- |
| | $\alpha_0$ | $\beta_0$ | $\alpha_1$ | $\beta_1$ | $\alpha_2$ | $\beta_2$ |
| KNN | 0.364341 | 1.133695 | 0.185119 | 2.393936 | 0.035711 | 2.12244 |
| XGBoost | 0.376170 | 1.138444 | 0.166671 | 2.383279 | 0.033399 | 2.117243 |
| MLP | 0.3870 | 1.131395 | 0.168273 | 2.368105 | 0.028425 | 2.120199 |

physiological pressure loadings, like 8 mm Hg applied in this study. To overcome this issue, a higher pressure loading may help to reduce the uncertainty of estimating $q_3$. For example, the Klotz curve can be helpful by providing extra pressure–volume relationship up to 30 mm Hg based on *ex vivo* experiments [36]. Indeed in a recent study, Lazarus *et al.* [63] have found that by including the Klotz curve as a prior when inferring the same set of myocardial parameters (**q**), the uncertainty in $q_3$ is reduced significantly.

A further challenge in UQ is the computational complexity when using the forward mathematical model. For example, the computational demand can be extremely high if Monte Carlo techniques are used for uncertainty propagation. For example, one run of our LV model takes 6 min, then 10 000 simulations will need 41 days. Thus, Monte Carlo methods using the LV model would be impractical. To overcome this, the surrogate model, a fast-running statistical approximation of the mathematical model allowing rapid prediction, can greatly reduce the computational burden and complexity, like the KNN, the XGBoost and the MLP studied here. They could potentially be used to make inferences about uncertainty in a LV biomechanical model. Another closely related task of UQ is sensitivity analysis for identifying important input parameters. Surrogate models have been used in sensitivity study of cardiac models [70] for alleviating the computational expense. Till now, most of UQ and sensitivity studies have been carried out on electrophysiology modelling [39,68,69] but less on biomechanics modelling of cardiac dynamics [70]. A comprehensive study of UQ in biomechanical cardiac models is necessary if they are aiming for clinical applications.

One limitation of the current approach is that all the data are generated from one single human LV, while LV function is different for different subjects, e.g. LV geometry, material property. To test whether the ML surrogate model is feasible for other subjects, we first extract two cases with different LV geometries from our previous study [35]. In a similar way, by assuming the parameters are known, we then simulate the FE LV model for generating the synthetic ground truth data, then the trained ML models are used to predict the output features using the same parameters. Table 6 is the MSE between the synthetic ground truth data for the two cases and the predictions from the ML models. The very large MSEs in table 6 clearly suggest that the LV geometry needs to be included as input variables. Methods are being developed for treating the geometry as input variables using dimension reduction techniques, such as principal component analysis (see discussion in [27]).

A further limitation is that the LV geometry was reconstructed from *in vivo* images, which is not stress free because of the non-zero blood pressure in the LV cavity; since in this study we do not intend to infer subject-specific material parameters using *in vivo* data, instead to compare the three ML models for learning generic LV dynamics in diastole and their suitability of inferring parameters by replacing expensive FE LV models. Future work should retrain the ML models with subject-specific LV models starting from the stress-free state for clinical applications, this will require tremendous efforts for building such a large number of personalized heart models, currently not available. Still, determining the fully stress-free state of the LV from *in vivo* data is very challenging because the heart is always pressurized. Future studies shall quantify how such a loaded geometry will affect the parameter estimation using *in vivo* data.

Last but not least, for each output feature, a sub-ML model is trained separately. It is expected the output features are correlated since they are generated from one LV model and each pair describes one set of data. Therefore a multi-output ML model would potentially reduce the prediction error and even the parameter uncertainty, as found in [26], in which a multivariate output Gaussian process has much less parameter estimation error compared with separate univariate output Gaussian processes. In fact, a multiple regression model can be readily developed with multi-output features using our data. Interestingly, our own preliminary test on the MLP model using three sub-models, each for one pair of ($\alpha_i$, $\beta_i$), shows that the mean $R^2$ score is actually marginally higher from the six sub-models

compared with the three sub-models (0.9994 vs 0.9993). Further work is needed to identify optimal schemes for learning LV dynamics in diastole, which is out of scope of this study.

# 5. Conclusion

In real-world optimization problems, the function evaluations usually require a large amount of computational time. Surrogate-based modelling and optimization play a valuable role in solving the large-scale computational problem. In this paper, we have developed three surrogate models and applied these methods to solve the parameter estimation problem of LV myocardium. Three surrogates based on different ML methods including the KNN, XGBoost and MLP are trained with 10 000 samples by simulating a computationally expensive finite-element ventricular model in diastole. All samples are generated using a Latin hypercube sampling method for a full coverage of the input parameter space. By comparing with a traditional gradient-based optimization method, our results show that the ML models can learn the relationships of pressure–volume and pressure–strain very well, and the parameter inference using the surrogate model can be carried out in minutes. In particular the XGBoost and MLP surrogate models have much less uncertainties in estimated parameters compared with the KNN model. Our results further suggest that the XGBoost surrogate model is the best one for predicting the LV diastolic dynamics and estimating parameters than the other two surrogate models. Further studies are warranted to investigate how the XGBoost surrogate model can be used for fast emulating cardiac pump function in a multi-physics and multi-scale framework.

Data accessibility. Data and relevant code for this research work are stored in GitHub: https://github.com/ ren2504413601/regression-surroages and have been archived within the Zenodo repository: https://doi.org/10. 5281/zenodo.4280222.

Authors' contributions. L.C. and L.R. contributed equally to this paper. L.R. and H.G. wrote the manuscript. L.R. performed machine learning models. L.C. and H.G. supervised the overall project. All authors analysed and reviewed the results.

Competing interests. We declare we have no competing interests.

Funding. This research is supported by the National Natural Science Foundation of China (grant nos. 11871399 and 11471261). G.Z. acknowledges the funding from the National Natural Science Foundation of China (grant no. 11802227). H.G. also acknowledges the funding from the Engineering and Physical Sciences Research Council (EPSRC) of the UK (grant nos. EP/N014642/1, EP/S030875/1 and EP/T017899/1).

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
