## [Reviewer comments · Royal Society Open Science]

Review History

RSOS-201121.R0 (Original submission)

Review form: Reviewer 1 (Chon Lok Lei)

Is the manuscript scientifically sound in its present form?

Yes

Are the interpretations and conclusions justified by the results?

Yes

Is the language acceptable?

Yes

Do you have any ethical concerns with this paper?

No

Have you any concerns about statistical analyses in this paper?

No

Recommendation?

Major revision is needed (please make suggestions in comments)

Comments to the Author(s)

Comments please see the attached file (Appendix A).

Review form: Reviewer 2

Is the manuscript scientifically sound in its present form?

Yes

Are the interpretations and conclusions justified by the results?

Yes

Is the language acceptable?

Yes

Do you have any ethical concerns with this paper?

No

Have you any concerns about statistical analyses in this paper?

No

Recommendation?

Accept with minor revision (please list in comments)

Comments to the Author(s)

Author developed three surrogate models based on three machine learning methods, the KNN, XGBoost and MLP to estimate the material parameters of left ventricle myocardium using the cavity volume and the maximum and minimum principal strains. This study is very novel and interesting addressing an important clinically relevant problem. Authors might have to discuss that in the real-world, the in vivo images are acquired with the physiological pressure, that is, the heart configuration obtained is pressurised. However, the mechanical analysis needs to be performed based on stress-free configuration, alternatively, at least based on the zero-pressure configuration. How this will affect the parameters estimation following the procedure as proposed in this study?

Minor comments

1. The full description should be given when the abbreviation firstly appeared in the main text.
2. It must be a typo for 'Figure 7' in the 2nd paragraph in the section (ii) strategy for output features.

Decision letter (RSOS-201121.R0)

Dear Mr Lei,

The Editors assigned to your paper RSOS-201121 "Surrogate models based on machine learning methods for parameter estimation of left ventricular myocardium" have now received comments from reviewers and would like you to revise the paper in accordance with the reviewer comments and any comments from the Editors. Please note this decision does not guarantee eventual acceptance.

Please submit your revised manuscript and required files (see below) no later than 21 days from today's (ie 23-Oct-2020) date. Note: the ScholarOne system will 'lock' if submission of the revision is attempted 21 or more days after the deadline. If you do not think you will be able to meet this deadline please contact the editorial office immediately.

on behalf of Dr Oliver Jensen (Associate Editor) and Mark Chaplain (Subject Editor)
openscience@royalsociety.org

Associate Editor Comments to Author (Dr Oliver Jensen):

Please revise your paper, responding in detail to all the points raised by the reviewers.

Reviewer comments to Author:
Reviewer: 1
Comments to the Author(s)

Comments please see the attached file.

Reviewer: 2

Comments to the Author(s)

Author developed three surrogate models based on three machine learning methods, the KNN, XGBoost and MLP to estimate the material parameters of left ventricle myocardium using the cavity volume and the maximum and minimum principal strains. This study is very novel and interesting addressing an important clinically relevant problem. Authors might have to discuss that in the real-world, the in vivo images are acquired with the physiological pressure, that is, the heart configuration obtained is pressurised. However, the mechanical analysis needs to be performed based on stress-free configuration, alternatively, at least based on the zero-pressure configuration. How this will affect the parameters estimation following the procedure as proposed in this study?

Minor comments

1. The full description should be given when the abbreviation firstly appeared in the main text.
2. It must be a typo for 'Figure 7' in the 2nd paragraph in the section (ii) strategy for output features.

===PREPARING YOUR MANUSCRIPT===

Your revised paper should include the changes requested by the referees and Editors of your manuscript. You should provide two versions of this manuscript and both versions must be provided in an editable format:
one version identifying all the changes that have been made (for instance, in coloured highlight, in bold text, or tracked changes);
a 'clean' version of the new manuscript that incorporates the changes made, but does not highlight them. This version will be used for typesetting if your manuscript is accepted.
Please ensure that any equations included in the paper are editable text and not embedded images.

===PREPARING YOUR REVISION IN SCHOLARONE===

To revise your manuscript, log into <https://mc.manuscriptcentral.com/rsos> and enter your Author Centre - this may be accessed by clicking on "Author" in the dark toolbar at the top of the

page (just below the journal name). You will find your manuscript listed under "Manuscripts with Decisions". Under "Actions", click on "Create a Revision".

Author's Response to Decision Letter for (RSOS-201121.R0)

See Appendix B.

RSOS-201121.R1 (Revision)

Review form: Reviewer 1 (Chon Lok Lei)

Is the manuscript scientifically sound in its present form?

Yes

Are the interpretations and conclusions justified by the results?

Yes

Is the language acceptable?

Yes

Do you have any ethical concerns with this paper?

No

Have you any concerns about statistical analyses in this paper?

No

Recommendation?

Accept as is

Comments to the Author(s)

I would like to thank the authors for addressing my concerns and comments thoroughly, and I have no further comments from a reviewing perspective.

Decision letter (RSOS-201121.R1)

This year has been very difficult for everyone, and we want to take the opportunity to thank you for your continued support in 2020.

The Royal Society Open Science editorial office will be closed from the evening of Friday 18 December 2020 until Monday 4 January 2021. We will not be responding during this time. If you have received a deadline within this time period, please contact us as soon as possible to allow us to extend the deadline. If you receive any automated messages during this time asking you to meet a deadline, we offer apologies and invite you to respond after the festive period or during normal working hours.

With our best for a peaceful festive period and New Year, and we look forward to working with you in 2021.

Dear Mr Lei,

It is a pleasure to accept your manuscript entitled "Surrogate models based on machine learning methods for parameter estimation of left ventricular myocardium" in its current form for publication in Royal Society Open Science.

on behalf of Dr Oliver Jensen (Associate Editor) and Mark Chaplain (Subject Editor)
openscience@royalsociety.org

Reviewer comments to Author:
Reviewer: 1

Comments to the Author(s)
I would like to thank the authors for addressing my concerns and comments thoroughly, and I have no further comments from a reviewing perspective.

Appendix A

Comments to manuscript “Surrogate models based on machine learning methods for parameter estimation of left ventricular myocardium”

Date: 28 September 2020

Summary

This manuscript has demonstrated using three classical machine learning regression techniques to build surrogate models for a left ventricular myocardium model. The three regression methods are the KNN, XGBoost, and MLP, aiming to model the relationship between a set of simplified model parameters of the Holzapfel-Ogden constitutive model and the parameters of an exponential function which describes the (end-diastolic) pressure-normalised volume, pressure-mean maximum principal strain, and pressure-mean minimum principal strain relationships. Each of the regression methods were able to emulate the simplified input-output relationship of the high-fidelity model well, in particular the XGBoost was shown to be the best amongst the three methods. The authors also showed the usefulness of these surrogate models for speeding up parameter inference.

GitHub repository

(Comments for the version on 2020-09-28, `commit:f6be4506764dbf73dfa8a638c5328ebe6cb75721`)

- Please provide an English translation for the README in the github repo.
- Please translate the Jupyter notebook to English.
- Please provide codes to allow replication of the latin hypercube sampling, rather than considering them as part of the “data”.
- From the Jupyter notebook’s In[9] and In[10], it seems like “outliers” were discarded which is not described/discussed in the manuscript.
- The Jupyter notebook was not fully run (it contains keyboard interruption!).
- Please provide a more detailed instruction for using the repository, and it’d be best if the repository contains enough code and instruction to reproduce the results in the manuscript.
- Please deposit the latest version to e.g. figshare or Zenodo as per the journal’s guideline.

Comments

(Comments are ordered according to the presentation of the study.)

Major

- Please make clear differences between the current study and those previous studies mentioned in the manuscript (i.e. refs [29], [30], [31], and [32] mentioned towards the end of the introduction section). Perhaps including more methods, such as Gaussian process, to perform a comparison/benchmark study would be more appropriate?
- Grid search and searching along axes might not be the best way to optimise hyperparameters, the authors may also consider methods such as Bayesian optimisation or those mentioned in their ref [52]. On this note, in Figure 5, although for low dimensional problem here one hidden layer for MLP models is usually sufficient, the authors may also try other architectures, i.e. choosing both numbers of layers and numbers of neurons?
- Please explain how the optimisations were initialised in Section 3.c, in particular please explain why the initialisation of the objective function between Figure 9A and B differed by 2 orders of magnitudes and/or why it is appropriate (as this may hinder the comparison of the “cost time” in Table 5).

- I would suggest, for Figure 10, the authors can discuss the fact that the similarity between B and D shows B is closer to the ground truth D. Also, please address why the surfaces in A and C are so different from D (and B), while their predictions in Figure 8 are much more closer to true values.
- The inference process was dramatically speeded up using the surrogate models, however the accuracy of inferred parameters using these surrogate models was inferior to that using the lsqnonlin solver with ABAQUS simulations (as shown in Table 5). Please quantify the uncertainty (or error) when using these surrogate models; this can be important when addressing the next comment (as well as the comment for Figure 10 below).
- Please discuss how the uncertainty in the emulators needs to be propagated when applying to real data. These should include discussion for the fact that the error between the surrogate model and the finite element model is not the only source of error in the estimates provided in the manuscript, but also the discrepancy between the finite element model (the ‘ground truth’ in this manuscript) and the reality; see e.g. <https://doi.org/10.1098/rsta.2019.0349> for potential issue that model discrepancy can cause during parameter estimation.
- Please discuss how surrogate models (with their computational speed up) can help in quantifying uncertainty and/or variability in multiscale modelling (see different levels of variability in e.g. <https://doi.org/10.1098/rsta.2019.0335>).

Minor

- page 18, line 40, please explain and justify the choice of $\mathbf{q} \in [0.1, 5]^4$.
- page 22, lines 49-59, please include a justification of using R-squared to evaluate the predictivity of the surrogate models.
- Please cite the software and/or describe the implementation of the surrogate models; from the GitHub repository provided, it seems like the authors used the `sklearn` and `xgboost` implementations which should be cited and clearly described.
- I would recommend moving Section 3(a) “Output feature reduction” to part of the methods section.
- page 28, lines 47-50: “Comparing with the conventional parameter estimation based on numerical evaluations of the forward simulation model, the ML-based surrogate models provide a significant improvement in computational time with an acceleration by tens or even hundreds of times.” Perhaps providing a comparison between the evaluation times of the finite-element model and of the surrogate models per forward simulation (i.e. time needed to compute an output given an input) would be more appropriate than comparing the time needed to do the inverse problem (as the authors seem to have initialised the optimisation differently and/or using different optimisation algorithms for these methods, see the comment above).
- page 28, lines 57-58, please include this as part of the motivation in the introduction or method sections.

Figures

- Figure 2, I’d recommend removing A-C, as they are shown in D-F already. I think it’s clear enough to just present D-F.
- Figure 3, I am not sure calling the arrow from “real world system” to “finite element model” as “simulation” is the right term, because one needs training for the finite element model which may contain discrepancy as well; it complicates the whole discussion/results presented in this manuscript.
- Figure 4, please explain clearly why it assumes or considers the prediction terms being independent. I’d expect some degree of correlation between the predicted outputs. And having the model to learn this kind of correlation could be crucial to the predictions. E.g. multi-input-multi-output model for the MLP? Please show why one is better than the other. There is an attempt to explain this in the Discussion section (lines 51-56), but I think it should take more that to justify the assumption.
- Figure 5, (1) in the caption, add “A schematic of a” before “multilayer perceptron model”. (2) It can be useful to provide similar schematics for the other two surrogate models (KNN and XGBoost).
- Figure 6, the caption is not clear, perhaps indicate “(i)” to “(iv)” in the figure as well. And missing a space between “function” and “f(q)” in the figure.
- Figure 7, the authors should explain why the samples are (seemingly?) skewed using the Latin hypercube

sampling. And perhaps plotting pairwise (co-variate) parameters would add more information to how samples are distributed; if not, violin plots may be useful here too. Also, maybe add a second y-axis (on the right) for the normalised values, for ease of comparison later (in Figure 8).

- Figure 8, please (1) rearrange the figure such that one row per model and one column per output features (or the other way round to fit the page size if needed); and (2) maybe combine Tables 3 and 4 with this figure, it'd be much easier to compare the plots and the MSE and R^2 scores in one place; (3) either change the caption or the plot such that they explain these are shifted and normalised values (Section 2.b.ii).
- Figure 9, I cannot see the ground truth on C and D, I suppose it's covered by the "lsqnonlin solver" line? Needs better way to visualise it. Missing space in y-axis label before the unit (kPa) for both C and D. Also please refer the "lsqnonlin solver" to as "ABAQUS simulation", as I suppose "lsqnonlin" is just a method solving nonlinear least-squares problems (which has never been mentioned in the main text)? And similarly for Table 5.
- Figure 10, (1) in the caption, perhaps reiterate what q_3 and q_4 are here? (2) Also please change "distribution" to "surface plot" (see below comment for page 27, line 19). (3) Would zooming into a smaller range reveal better the contour? Or maybe try plotting the logarithm of the contour values? It hardly shows anything changing within this range, and this does not explain the shift in the estimated parameters in Table 5 — the colours of the obtained (shifted) parameters and the true values are the same. And quantifying the uncertainty may help explaining this too — is it a systematic shift (i.e. error of the emulators) or an unidentifiability issue (i.e. the objective function is 'flat')?

Cosmetic

- page 14, abstract, spell out "LV" as left ventricular.
- page 15, line 13, replace "i.e." with "e.g."
- page 15, line 14, "`\textit{in silico}` medicine" instead of "sillico medicine".
- page 15, line 27: "In general, material parameter estimation of a FEM heart model is formulated as an inverse problem [14–18], which essentially requires solving a constrained optimization problem [19,20] by..." the authors have excluded many other approaches for solving an inverse problem, I suppose changing "which essentially requires solving" to something like "for example it requires solving" would be more appropriate.
- page 15, line 30, again "generally" may not be an appropriate word, as gradient-based methods are one of the many available methods for solving optimisation problems.
- page 15, line 31, (1) insert "by" before "some intelligent methods"; (2) maybe add "(also known as nature-inspired methods)"; (3) replace "i.e." with "e.g." (as what follow are some examples); (4) it's more commonly known as "genetic algorithms" than "genetic method" (and it's a class of methods).
- page 15, line 36, I'd refer the model to as something like "high-fidelity" instead of "high-accuracy" (a more complex model is by no means more accurate).
- page 15, line 43, I'd recommend mentioning surrogate models are (recently) used for the cardiac modelling (ranging from cell level to organ level) here (before the next paragraph where the authors give more examples).
- page 15, line 44, please describe the changes here. Maybe "A recent research" rather than "The latest research" when referring to a publication in 8 years ago...
- page 15, line 44-47, please rewrite.
- page 15, line 46, remove "intensive", and please rewrite "ranging from ... to ..." and provide references as examples.
- page 15, line 46, please define machine learning as "ML", as it is the first appearing in the main text.
- page 15, line 48, please write left ventricular next to "LV" as the authors first mention it in the main text.
- page 15, line 56, maybe also mention <http://dx.doi.org/10.1098/rsta.2019.0334>?
- Figure 1 caption, "Visualisation" instead of "Visualise".
- page 16, line 56, define Ψ as the strain-energy function in the text.
- page 17, line 43, misplaced period, "Klotz et al."

- page 17, line 48, insert “and” before “ $\bar{\epsilon}_{\max}$ ”.
- page 20, Eq. (2.8), the “ $L_{1,2}$ ” notation seems confusing, maybe just “L”?
- page 20, line 41, L1, L2 missing subscripting, and either “a distance function” or “a metric”, but not “distance metric function”.
- page 20, Algorithm 1, I don’t find the pseudo-algorithm adds more information than the text has already explained; maybe remove and just add a sentence before this to clarify the hyperparameters are K and the distance function.
- page 21, line 9, “Chen and Guestrin [40]” (missing the second author).
- page 21, line 10, remove “engineering”, and spell out “GBDT”.
- page 21, line 12, spell out “CART”. And “CART tree” is a redundant acronym, just CART is fine (T stands for tree).
- page 21, line 25, again, either “a distance function” or “a metric”.
- page 21, line 26, misplaced “(t)” in the target data, that should go to the first y.
- page 23, line 10, please refer the size of hidden layer to as “number of neurons” as well for clarity.
- page 23, Table 1, please provide the detail of the software packages used before referring to the specific function arguments of that particular software!
- page 23, line 53, capitalise “Python”.
- page 27, line 8, “. . . using the inverse approach developed. . . ”.
- page 27, line 19, I think it’s better to call the plots in Figure 10 as “surface plots of the objective function” instead of “distributions of objective function”, as it is not a distribution in a statistical sense, which can be confusing.
- page 27, line 22, missing mention of the lsqnonlin solver/ABAQUS simulation.
- page 27, line 27, suggest referring to Eq. 2.5 here, i.e. “. . . which may be explained by the limitation of the reduced material parameter space in Equation (2.5).”
- page 29, line 15, missing period, “Gao et al.”.

Appendix B

Responses to reviewers for the manuscript “Surrogate models based on machine learning methods for parameter estimation of left ventricular myocardium”

We would like to thank the reviewers for their constructive comments. In the following, we provide point-to-point responses. The original comments are in italic. All the corrections in the manuscript are highlighted with blue color.

Referees: 1

GitHub repository

We have checked and corrected all the codes according to your valuable advice. Now we create a new GitHub repository, which is all written in English. It contains the source code for training the machine learning (ML) models, Latin hypercube sampling and parameter estimation. We add some documentations in all modules. More details can be found in

<https://github.com/ren2504413601/regression-surroages>

The former repository using Jupyter notebook is organized into “train” and “optimization” folders. We now add a folder named “sampling”, it contains the codes for implementing the Latin hypercube sampling. In the previous version, we tried to remove some outliers for training ML models, while this did not improve the performance. Now we do not remove outliers, all data are used for training and testing ML models.

Major

Q1: Please make clear differences between the current study and those previous studies mentioned in the manuscript (i.e. refs [29], [30], [31], and [32] mentioned towards the end of the introduction section). Perhaps including more methods, such as Gaussian process, to perform a comparison/benchmark study would be more appropriate?

Response:

Thanks for the very constructive suggestion. In the introduction section, we now review the main works of previous studies in more detail, and point out the innovation and difference of our current work. Please see the last two paragraphs of introduction. (page 2 line 40-page 3 line 72). It reads:

Recent studies have shown that myocardial property could be a potential biomarker of predicting ventricular pump function recovery post-myocardial infarction [1]. Estimation of myocardial material parameters from image-based models has attracted intensive interests by formulating a gradient-based inverse problem [2] or using machine learning (ML)-based surrogate approaches for fast parameter inferences [3,4]. By using ML models, the behaviors of the left ventricular (LV) in response to changes in material properties, loads and boundary conditions etc. can be predicted in real time, and can be further applied to the design of medical instruments and monitoring heart condition [5]. For instance, Liang et al. [6] has been first put forward the deep learning technique, a multilayer neural network, as a surrogate of FEM for stress analysis, and the trained model was capable of predicting the stress distributions of aorta

by replacing the complex structural finite-element analysis with an average error of $< 1\%$ for ML predicted stress distribution. Dabiri et al. [5] adopted eXtreme Gradient Boosting (XGBoost) and Cubist to predict the LV pressures, volumes as well as LV stresses by training them with hundreds of forward FEM simulations of a biomechanical LV model, and their results showed that the surrogate ML models can predict LV mechanics very accurately and are much faster than the FEM models. But model calibration using the ML-based surrogate model has not been carried out in these two studies [5,6]. Achille et al. [7] inferred the unloading LV geometry used Gaussian process and further statistically learned the infarct shape and size on LV performance in patients extracted from a public database [8]. More recently, Longobardi et al. [9] predicted left ventricular contractile function via Gaussian process emulation in aortic-banded rats, the Bayesian history matching was applied to constrain the initial parameter sets in order to exclude those points which generate non-physiological biomechanical models. They further performed a Sobol sensitivity analysis using the trained emulator. Most of studies have been focused on demonstrating the feasibility, reliability and accuracy of emulating the cardiac models with various ML approaches, and only a few studies have investigated how the inference problem of unknown parameters can be accelerated using ML surrogate models. For example, Noe and Davies et al. [10,4] both have presented a statistical emulation framework for emulating LV mechanics using Gaussian process aiming for accelerating the parameter estimation of myocardium from in vivo data. Tens of thousands of simulations of a LV biomechanical model have been performed to train their statistical emulation framework. In both studies, LV cavity volume and 24 circumferential strains at end-diastole were predicted from the trained emulator, but not the dynamics in diastole. Both studies have demonstrated that the computational costs can be reduced by about three orders of magnitude.

We also train a Gaussian process model using the same data. Its comparison and discussion can be found in the second paragraph in the Discussion section (page 17, line 396 - 408). It reads:

Gaussian process also has been applied to parameter estimation in myocardium [10,4] using a synthetic LV model, while their output features were only from the measurements at end-diastolic phase but not for the whole diastole. In a followed study, Lazarus et al. [11] found that because of lack data, only using end-diastolic measurements makes it challenging to infer q_3 and q_4 . Instead, in this study the parameterized pressure–volume and pressure–strain data in diastole are used for training the ML models and parameter inferences. Still we observe large uncertainties associating with q_3 and q_4 . Thus extra data will be needed for identifying q_3 and q_4 with high confidence. We also have trained a Gaussian process model for our data, and the prior covariances of Gaussian process are computed by using a Radial-basis kernel function. We then use the trained Gaussian process model for inferring \mathbf{q} , the inferred \mathbf{q} is (1.0053, 1.0027, 1.0812, 1.1504) with $\mathbf{f}(\mathbf{q}) = 1.7424e^{-5}$. It is slightly poorer than the XGBoost model as seen in Table 4. While this should still suggest that Gaussian process is a good surrogate model as XGBoost for learning LV dynamics in diastole.

Q2: *Grid search and searching along axes might not be the best way to optimise hyper-parameters, the authors may also consider methods such as Bayesian optimisation or those mentioned in their ref [52]. On this note, in Figure 5, although for low dimensional problem here one hidden layer for MLP models*

is usually sufficient, the authors may also try other architectures, i.e. choosing both numbers of layers and numbers of neurons?

Response:

We now discuss various methods of hyper-parameter optimization in Discussion section, and a preliminary comparison using Bayesian optimization method is also included. (page 17 line 419 – page 18 427).

There are various methods for hyper-parameter tuning [12], including random search, grid search, Bayesian optimization, etc. Random search uses a randomized search over parameters in which each setting is sampled from a distribution over possible parameter values. Grid search generates candidates from a grid of parameter values. Bayesian optimization [13] is a global optimization method based on sequential training of a statistical approximation (Gaussian process) of the target function. We also apply the Bayesian optimization for hyper-parameter tuning using bayesopt [14] implemented in Python. By comparing with the trained ML models using the grid search approach, the R^2 scores are almost the same, while the Bayesian optimization takes much longer time for tuning ML models than the grid search approach.

We now have trained a two hidden layer neural network, the results are almost the same as the one hidden layer neural network. (page 17, line 409 – 418)

In a recent study, Liu et al. [15] has used a two hidden-layer neural network for estimating in vivo constitutive parameters of aortic wall. Considering the LV dynamics in diastole is a passive expansion process with increased volume and strain magnitude as shown in Figure 2, a deep neural network may over-fit our data easily because of its large number of freedom. As a test, we implement a two hidden-layer neural network and train using the same data with a fixing learning rate of $1e^{-3}$, and the grid search is used for selecting optimal neuron numbers at each hidden layer with the grid values in Table 1. Our result shows that the mean R^2 score for the testing data is almost the same as using the one hidden layer neural network, which may suggest that the one-layer MLP can learn the LV FE model in diastole well. For a much complex dynamics system, i.e. whole heart contraction, a deep neural network would be necessary.

Q3: Please explain how the optimisations were initialised in Section 3.c, in particular please explain why the initialisation of the objective function between Figure 9A and B differed by 2 orders of magnitudes and/or why it is appropriate (as this may hinder the comparison of the “cost time” in Table

Response:

The optimization using ABAQUS simulation is implemented in Matlab using lsqnonlin function which uses a trust-region-reflective algorithm. The optimization using ML-based surrogate models is implemented using a differential evolution algorithm from a Python third part library scipy.optimize.

The initialization of lsqnonlin solver and differential evolution algorithm are all randomly selected from the range $[0.1, 5]^4$. To our best knowledge, the initial objective function of the differential evolution algorithm is the minimal function value of initial candidates, the population size is set to 50 in our study. The default population initialization is done by the Latin Hypercube sampling (SciPy (version 1.0.0) documentation), which can maximize coverage of the available parameter space, thus the objective function in the first step for the ML models can be much smaller than the optimization using the lsqnonlin

solver.

We now update the Parameter inference section (page 11 line 286-291, 303-306; page 12 line 311-314). It reads:

Differential evolution is a stochastic population based method which has been applied for global optimisation problems. The optimization procedure is implemented using the Python library `scipy.optimize`. We choose the total population size is 50, the relative tolerance for convergence is $1e^{-6}$, the maximum number of iterations is 500, the mutation constant is in (0.5, 1), the recombination constant is 0.7, and initial values are randomly selected in the range $[0.1, 5]^4$. (SciPy (version 1.0.0) documentation)

Trust-region-reflective is a gradient-based algorithm with fast convergence but it may not guarantee a global optimal. We set the maximum number of iterations 500, the relative tolerance on the target value $1e^{-6}$, and initial values are also randomly selected from the range $[0.1, 5]^4$.

All the computations are carried out in a 64-bit Windows 7 workstation with 1.7 GHz Intel Core E5-2609 CPU and 32 GB RAM. ABAQUS simulation and training ML models are performed on sixteen threadings for parallelism. No parallelization is used when inferring material parameters within `lsqnonlin` and differential evolution solvers.

Q4: *I would suggest, for Figure 10, the authors can discuss the fact that the similarity between B and D shows B is closer to the ground truth D. Also, please address why the surfaces in A and C are so different from D (and B), while their predictions in Figure 8 are much more closer to true values.*

Response:

The figure now is updated with the surface plots of logarithm objective function (Figure 10, page 15). The description of the figure is also updated (page 15 line 357 - 363). It reads:

It can be found that the surface plots of using the XGBoost, MLP and ABAQUS simulation are similar, the minimum from the ABAQUS simulation is clearly identifiable, but a narrow valley parallel to q_4 with multiple local minimums for the XGBoost, and that valley becomes a flat region for the MLP model. Thus, q_4 seems unidentifiable when using the XGBoost and MLP models even with fixed $q_1 = 1$ and $q_2 = 1$. The surface plot of the objective function for the KNN model is very different from the others, which again indicates its poor performance of learning the FE LV model.

Q5: *The inference process was dramatically speeded up using the surrogate models, however the accuracy of inferred parameters using these surrogate models was inferior to that using the `lsqnonlin` solver with ABAQUS simulations (as shown in Table 5). Please quantify the uncertainty (or error) when using these surrogate models; this can be important when addressing the next comment (as well as the comment for Figure 10 below).*

Response:

We add the uncertainty quantification for parameter inference by using a bootstrap method, the last paragraph of subsection “parameter inference” (page 16 line 374-386), and further discussed in the Discussion section (page 18, line 449-461). It reads:

In order to have an indication of the uncertainty in our inference using the three ML models, we adopt a bootstrap approach [11] for the inverse uncertainty quantification. We first compute the prediction value y^{pred} using the estimated parameters in Table 4, we then compute the

residuals between the referential values y^{ref} (i.e. the output features which are obtained from the synthetic LV model), they are $\epsilon : \{ \epsilon_i = y_i^{ref} - y_i^{pred}, (i = 1,2,3,4,5,6) \}$. By randomly selecting all combinations from ϵ , and 0 used from unselected residuals as supplement, we can obtain a set of residuals $\mathcal{R} = \{ \hat{\epsilon}_j \}_{1 \leq j \leq 64}$, and then generate surrogate data $\hat{y}_j = y^{pred} + \hat{\epsilon}_j$, where $\hat{\epsilon}_j$ is the J-th draw from \mathcal{R} . We then repeat the parameter inference on each \hat{y}_j to obtain new estimations. The mean and standard deviation of all estimated parameters and objective function values are listed in Table 5. The uncertainties associating with q_1 and q_2 are much less than the uncertainties for q_3 and q_4 , again the XGBoost has smaller uncertainties in q_3 and q_4 compared to the KNN and the MLP model. The large uncertainties associating with the KNN model suggests its unsuitability for inferring constitutive parameters by replacing a FE LV model.

As shown in Table 4, estimated parameters using those three trained surrogate models do not match the ground truth values, and also inferior to the values inferred directly using the forward ABAQUS LV model. To quantify the associated uncertainty in inferred parameters, we further adopt a bootstrap approach, results are summarized in Table 5. In general q_1 and q_2 can be inferred with high accuracy, but not q_3 and q_4 , especially for the KNN. XGBoost achieves the best performance with very little uncertainties in q_1 , q_2 and q_4 , but poor in q_3 , that is partially because q_3 is related to nonlinear response of collagen network and myofibres [16], which are not fully engaged under normal physiological pressure loadings, like 8 mmHg applied in this study. To overcome this issue, a higher pressure loading may help to reduce the uncertainty of estimating q_3 . For example, the Klotz curve can be helpful by providing extra pressure-volume relationship up to 30 mmHg based on ex vivo experiments [17]. Indeed in a recent study, Lazarus et al. [11] have found that by including the Klotz curve as a prior when inferring the same set of myocardial parameters (\mathbf{q}), the uncertainty in q_3 is reduced significantly.

Q6: *Please discuss how the uncertainty in the emulators needs to be propagated when applying to real data. These should include discussion for the fact that the error between the surrogate model and the finite element model is not the only source of error in the estimates provided in the manuscript, but also the discrepancy between the finite element model (the ‘ground truth’ in this manuscript) and the reality; see e.g. <https://doi.org/10.1098/rsta.2019.0349> for potential issue that model discrepancy can cause during parameter estimation.*

Response:

We now discuss the general uncertainty quantification (UQ) for applying mathematical models for decision making in Discussion section (page 18 line 428 – 448). It reads:

Uncertainty quantification (UQ) is an essential step in applying mathematical models for decision making, the LV model included. There are various sources of uncertainty in cardiac models, including intrinsic variability (i.e. collagen content), parameter uncertainty (i.e. myocyte stiffness), initial/boundary condition uncertainty (i.e. ventricular cavity pressure, pericardium), geometry uncertainty (i.e. myocyte architecture), model uncertainty (i.e. continuum approximation of soft tissue), etc. Interested readers refer to [18,19] for recent reviews of uncertainty and variability in cardiac models. If a surrogate model is used for prediction, an additional source of uncertainty is the model discrepancy between the surrogate model and the mathematical model. As evidenced in Figure. 8, the KNN has the worst

prediction, which may potentially explain the large discrepancy and high uncertainty in estimated parameters (Table 5). Optimal computational design strategy would be needed for achieve high accuracy for the surrogate model, for example sampling strategies according to the underline physics, dense coverage for small parameter values. In a recent study, Lei et al. [20] studied the model discrepancy between the mathematical model and reality in cardiac ionic models. They modelled the discrepancy using Gaussian processes and autoregressive-moving-average model for improved prediction of unseen situations, and found that different methods for accounting model discrepancy are needed for different models in different situations. The similar framework could also be applied to the finite element LV model, which is a numerical approximation of LV dynamics under various assumptions (i.e. homogenous material property, rule-based myofibre structure, ignoring viscous effects).

Q7: *Please discuss how surrogate models (with their computational speed up) can help in quantifying uncertainty and/or variability in multiscale modelling (see different levels of variability in e.g. <https://doi.org/10.1098/rsta.2019.0335>).*

Response:

We now discuss the related tasks including UQ and sensitivity study using surrogate models at page 18 line 462 – 475. It reads:

A further challenging in uncertainty quantification is the computational complexity when using the mathematical model. For example, the computational demand can be extremely high if Monte Carlo techniques are used for uncertainty propagation. For example, one run of our LV model takes 6 mins, then 10 thousand simulations will need 41 days. Thus Monte Carlo methods using the LV model would be impractical. To overcome this, the surrogate model, a fast-running statistical approximation of the mathematical model allowing rapid prediction, can greatly reduce the computational burden and complexity, like the KNN, the XGBoost and the MLP studied here, which could potentially be used to make inferences about uncertainty in a LV biomechanical model. Another closely related task of UQ is sensitivity analysis for identifying important input parameters. Surrogate models have also been used in sensitivity study of cardiac models [20] for alleviating the computational expense. Till now, most of UQ and sensitivity studies have been carried out on electrophysiology modelling [17,18,19] but less on biomechanics modelling of cardiac dynamics [21]. A comprehensive study of UQ in biomechanical cardiac models is necessary if they aim for clinical applications.

Minor

Q8: *page 18, line 40, please explain and justify the choice of \mathbf{q} in $[0.1, 5]^4$.*

Response:

The H-O constitutive in Equation (2.1) contains eight positive parameters. Many studies have shown that to estimate those parameters accurately is challenging. Thus we follow the parameterization from the previous studies by Noe and Dave et al [10,4]. The range of \mathbf{q} is adopted from these two studies, based on our previous study on estimating myocardial stiffness from 27 healthy volunteers. We now make it clear in for the choice of \mathbf{q} at page 5, line 134-133

The range of \mathbf{q} is adopted from [10,4] which was derived from the population average values reported in [22].

Q9: page 22, lines 49-59, please include a justification of using R-squared to evaluate the predictivity of the surrogate models.

Response:

The three chosen ML models can be considered to be supervised learning regression problem. In regression, the R-squared (coefficient of determination) is a statistical measure of how well the regression predictions approximate the real data points. Higher the R^2 , the better the model predictability. A score of 1.0 indicates a perfect prediction. By taking mean square error (MSE) as loss function, R^2 for evaluating test data, Dabiri et al [5] recently have trained ML models to predict the LV mechanics. Their R^2 scores are from 0.94 to 0.99 and the trained ML models can predict LV behavior very well. Like Dabiri et al's work [5], we choose MSE as the loss function for training ML model and use R^2 for evaluating test data. Our results show that the generalization of ML models seems very good. In addition, the mean squared error, mean absolute error and others can also be used to evaluate the predictability of a regression model. We now rewrite the description of R^2 at page 10, line 253-258

We further introduce coefficient of determination R^2 , which is used to evaluate the predictability of a ML model for choosing the optimal set of hyper-parameters similar as in [5],

$$R^2(y, \hat{y}) = 1 - \frac{\sum_{i=1}^n (y^i - \hat{y}^i)^2}{\sum_{i=1}^n (y^i - \bar{y})^2}, \quad \text{and } \bar{y} = \frac{1}{n} \sum_{i=1}^n y^i.$$

R^2 represents the proportion of variance that has been explained by input parameters in a model. A score of 1.0 indicates a perfect prediction, and it may be negative because a model can be arbitrarily worse, see Scikit-learn 0.23.2 documentation [23].

Q10: Please cite the software and/or describe the implementation of the surrogate models; from the GitHub repository provided, it seems like the authors used the sklearn and xgboost implementations which should be cited and clearly described.

Response:

Thanks for your suggestion. Implementation details are added in Section 2(C). (page 10 242-248. 266-268).

The KNN and MLP models are implemented using Scikit-learn library [23]. Specifically, the KNN model uses a KDTree algorithm [24] (a fast algorithm to generalise N-point problems) to compute the nearest neighbours. The MLP model uses an Adam solver [25] (a stochastic gradient-based optimizer) to optimize the MSE with L_2 penalty, see Equation (2.16). Early stopping is adopted to terminate training when validation score is not improved after 10 epochs. The maximum number of iterations is set to 200. The XGBoost model is implemented using the XGBoost Python library. The tolerance for the optimization is $1e^{-4}$.

All pre-defined grid values for each hyper-parameter are shown in Table 1, and the name of hyper-parameter follows the conventions in sklearn and XGBoost libraries.

Q11: I would recommend moving Section 3(a) "Output feature reduction" to part of the methods section.

Response:

Thanks for the insightful suggestion. The content for "Output feature reduction" is merged to subsection

“Strategy for output features” in Methods section, see page 6.

Q12: page 28, lines 47-50: “Comparing with the conventional parameter estimation based on numerical evaluations of the forward simulation model, the ML-based surrogate models provide a significant improvement in computational time with an acceleration by tens or even hundreds of times.” Perhaps providing a comparison between the evaluation times of the finite-element model and of the surrogate models per forward simulation (i.e. time needed to compute an output given an input) would be more appropriate than comparing the time needed to do the inverse problem (as the authors seem to have initialised the optimisation differently and/or using different optimisation algorithms for these methods, see the comment above).

Response:

We add one column named “Evaluation numbers” in Table 3 (page 14), and another column for the time cost of one run. It reads: (page 14 line 331-334)

For each prediction, the KNN model takes 0.08 s, and 0.02 s for the XGBoost, and 0.005 s for the MLP model, all are significantly less than the forward ABAQUS simulation, which takes around 6 minutes for one run. The computational time using the ML models is almost reduced by 3 to 4 orders, and summarized in Table 3.

Q13: page 28, lines 57-58, please include this as part of the motivation in the introduction or method sections.

Response:

The content is moved to the Methods section, see page 7 line 182- page 8 186. It reads:

The reasons of choosing these three ML methods are (1) the samples in this work are limited with a few input and output features, (2) the functional form shown in Figure 2 is relatively simple, which may suggest that simple ML models can be used, and (3) these selected ML models are easy to implement and readily available in various open-source packages, but they have not applied in LV model calibration.

Figures

Q14: Figure 2, I’d recommend removing A-C, as they are shown in D-F already. I think it’s clear enough to just present D-F.

Response:

Thanks for your suggestion. Figure (A-C) has been removed and merged with (E-F). The lasted version was showed as Figure 2 (page 5).

Q15: Figure 3, I am not sure calling the arrow from “real world system” to “finite element model” as “simulation” is the right term, because one needs training for the finite element model which may contain discrepancy as well; it complicates the whole discussion/results presented in this manuscript.

Response:

Yes, the ABAQUS simulation of the finite element LV model contains discrepancy. We update the emulation process from surrogate model to FE model in Figure 3 (page 6) and update the caption describing the fact that finite element method is used to approximate the real world system through simulation.

Q16: Figure 4, please explain clearly why it assumes or considers the prediction terms being independent. I'd expect some degree of correlation between the predicted outputs. And having the model to learn this kind of correlation could be crucial to the predictions. E.g. multi-input-multi-output model for the MLP? Please show why one is better than the other. There is an attempt to explain this in the Discussion section (lines 51-56), but I think it should take more that to justify the assumption.

Response:

The output features are ordered as $y = (y_1, y_2, y_3, y_4, y_5, y_6) = (\alpha_0, \beta_0, \alpha_1, \beta_1, \alpha_2, \beta_2)$. Here (α_i, β_i) are three pair of positive value derived by curve fitting. We have tried to learn the output features with three sub-models for the three pair of (α_i, β_i) . We find the mean R^2 score of 6 sub-models is marginally smaller than the ML model with three sub-models. So we adopt the strategy of 6 sub-models in this study.

We now discuss this in the last paragraph of Discussion section, page 19 line 497-507, it reads

Last but not the least, for each output feature, a sub-ML model is trained separately. It is expected the output features are correlated since they are generated from one LV model and each pair describes one set of data, therefore a multi-output ML model would potentially reduce the prediction error and even the parameter uncertainty, as found in [10], in which a multivariate output Gaussian process has much less parameter estimation error compared to separate univariate output Gaussian processes. In fact, a multiple regression model can be readily developed with multi-output features using our data. Interestingly, our own preliminary test on the MLP model using three sub-models, each for one pair of (α_i, β_i) , shows that the mean R^2 score is actually marginally higher from the 6 sub-models compared to the three sub-models (0.9994 v.s. 0.9993). Further work is needed to identify optimal schemes for learning LV dynamics in diastole, which is out of scope of this study.

Q17: Figure 5, (1) in the caption, add "A schematic of a" before "multilayer perceptron model". (2) It can be useful to provide similar schematics for the other two surrogate models (KNN and XGBoost).

Response:

We update the caption of this figure, now it is shown in Figure 6 (page 9).

Q18: Figure 6, the caption is not clear, perhaps indicate "(i)" to "(iv)" in the figure as well. And missing a space between "function" and "f(q)" in the figure.

Response:

We have labeled the iteration process in a new picture to replace the old picture and checked all formats, and the caption is updated. See Figure 7(page 12).

Q19: Figure 7, the authors should explain why the samples are (seemingly?) skewed using the Latin hypercube sampling. And perhaps plotting pairwise (co-variate) parameters would add more information to how samples are distributed; if not, violin plots may be useful here too. Also, maybe add a second y-axis (on the right) for the noramlised values, for ease of comparison later (in Figure 8).

Response:

We add violin plots to show the distribution of samples and their normalized values, which are shown in Figure 4 (page 7). The samples look skewing which is because the LV model is non-linear, even though the parameters \mathbf{q} are sampled using Latin Hypercube, this will not ensure the output features \mathbf{y} will have a symmetric distribution. We now make it clear at page 6 under subsection "Strategy for learning

output features”.

Q20: *Figure 8, please (1) rearrange the figure such that one row per model and one column per output features (or the other way round to fit the page size if needed); and (2) maybe combine Tables 3 and 4 with this figure, it'd be much easier to compare the plots and the MSE and R^2 scores in one place; (3) either change the caption or the plot such that they explain these are shifted and normalised values (Section 2.b.ii).*

Response:

We rearrange the figure follow the reviewer's suggestion that one column shows one ML model, and each column has six subfigures to show the prediction of six output features. We also merge Table 3 and Table 4 as annotations in the upper left corner of all subfigures. The caption is changed to indicate that output features are normalized values. All changes can be seen in Figure 8 (page 13).

Q21: *Figure 9, I cannot see the ground truth on C and D, I suppose it's covered by the “lsqnonlin solver” line? Needs better way to visualise it. Missing space in y-axis label before the unit (kPa) for both C and D. Also please refer the “lsqnonlin solver” to as “ABAQUS simulation”, as I suppose “lsqnonlin” is just a method solving nonlinear least-squares problems (which has never been mentioned in the main text)? And similarly for Table 5.*

Response:

We have checked and corrected all the mistakes there. “lsqnonlin solver” change to “ABAQUS simulation”, and also add space in y-axis label before the unit (kPa). To clearly show the myofibre stress–stretch relationships, we further compute the residual Cauchy stress by subtracting ground truth data with other four methods, which can be seen in Figure 11 (page 16). The descriptions read:

Figure 11(a)-(b) shows the myofibre stress–stretch relationship under uni-axial stretching mode derived from the H-O law (Equation (2.1)) with the parameters in Table 4. For the stretch along fibre direction, the inferred stress-stretch data from the three ML model and ABAQUS simulation are very close to the ground truth data. But larger discrepancies can be found for the Cauchy stress along the sheet direction when inferred from the three ML models. Figure 11(c)-(d) shows corresponding stress residuals, which are calculated by subtracting the inferred Cauchy stress from the ground truth data. It is not surprising that the inference using ABAQUS simulations achieves the best accuracy because the synthetic data are generated using the ABAQUS simulation. The XGBoost comes second close to the ground truth data with smaller residuals compared to the KNN and MLP models.

Q22: *Figure 10, (1) in the caption, perhaps reiterate what q_3 and q_4 are here? (2) Also please change “distribution” to “surface plot” (see below comment for page 27, line 19). (3) Would zooming into a smaller range reveal better the contour? Or maybe try plotting the logarithm of the contour values? It hardly shows anything changing within this range, and this does not explain the shift in the estimated parameters in Table 5 — the colours of the obtained (shifted) parameters and the true values are the same. And quantifying the uncertainty may help explaining this too — is it a systematic shift (i.e. error of the emulators) or an unidentifiability issue (i.e. the objective function is ‘flat’)?*

Response:

We plot the contours of logarithmic (to the base 10) objective function by fixing q_1, q_2 , while varying q_3, q_4 , which is shown in Figure 10 (page 15). There seems to be an unidentifiability issue for q_3 and

q_4 . Analysis of Figure 10 can be found at page 15 line 357-363. It reads:

It can be found that the surface plots of using the XGBoost, MLP and ABAQUS simulation are similar, the minimum location from the ABAQUS simulation is clearly identifiable, but a narrow valley parallel to q_4 with multiple local minimums for the XGBoost, and such narrow valley becomes a flat region for the MLP model. Thus, q_4 could be unidentifiable when the XGBoost and MLP models even with fixed $q_1 = 1$ and $q_2 = 1$. The surface plot of the objective function for the KNN model is very different from the others, which again indicates its poor performance of learning the FE LV model.

Uncertainty quantification is also added by using a bootstrap approach for the inverse uncertainty quantification. Extensive discussion can be found at page 18 line 449-363. It reads:

As shown in Table 4, estimated parameters using those three trained surrogate models do not match the ground truth values, and also inferior to the values inferred directly using the forward ABAQUS LV model. To quantify the associated uncertainty in inferred parameters, we further adopt a bootstrap approach, results are summarized in Table 5. In general q_1 and q_2 can be inferred with high accuracy, but not q_3 and q_4 , especially for the KNN. XGBoost achieves the best performance with very little uncertainties in q_1 , q_2 and q_4 , but poor in q_3 , that is partially because q_3 is related to nonlinear response of collagen network and myofibres [24], which are not fully engaged under normal physiological pressure loadings, like 8 mmHg applied in this study. To overcome this issue, a higher pressure loading may help to reduce the uncertainty of estimating q_3 . For example, the Klotz curve can be helpful by providing extra pressure-volume relationship up to 30 mmHg based on ex vivo experiments [25]. Indeed in a recent study, Lazarus et al. [11] have found that by including the Klotz curve as a prior when inferring the same set of myocardial parameters (\mathbf{q}), the uncertainty in q_3 is reduced significantly.

Cosmetic

Page 14, abstract, spell out “LV” as left ventricular.

Corrected.

Page 15, line 13, replace “i.e.” with “e.g.”.

Corrected.

Page 15, line 14, “*in silico* medicine” instead of “*silico* medicine”.

Corrected.

Page 15, line 27: “In general, material parameter estimation of a FEM heart model is formulated as an inverse problem [14–18], which essentially requires solving a constrained optimization problem [19,21] by. . .” the authors have excluded many other approaches for solving an inverse problem, I suppose changing “which essentially requires solving” to something like “for example it requires solving” would be more appropriate.

Corrected.

Page 15, line 30, again “generally” may not be an appropriate word, as gradient-based methods are

one of the many available methods for solving optimisation problems.

Corrected. Already replace “generally” with “for example”.

Page 15, line 31, (1) insert “by” before “some intelligent methods”; (2) maybe add “(also known as nature-inspired methods)”; (3) replace “i.e.” with “e.g.” (as what follow are some examples); (4) it’s more commonly known as “genetic algorithms” than “genetic method” (and it’s a class of methods).

Corrected.

Page 15, line 36, I’d refer the model to as something like “high-fidelity” instead of “high-accuracy” (a more complex model is by no means more accurate).

Corrected.

page 15, line 43, I’d recommend mentioning surrogate models are (recently) used for the cardiac modelling (ranging from cell level to organ level) here (before the next paragraph where the authors give more examples).

Corrected. We have rewritten the surrogate model in Introduction.

page 15, line 44, please describe the changes here. Maybe “A recent research” rather than “The latest research” when referring to a publication in 8 years ago. . .

Corrected.

page 15, line 44-47, please rewrite.

We have rewritten this following the Reviewer’s advice (page 2 line 40-44), it reads:

Recent studies have shown that myocardial property could be a potential biomarker of predicting ventricular pump function recovery post-myocardial infarction [1]. Estimation of myocardial material parameters from image-based models has attracted intensive interests by formulating a gradient-based inverse problem [2] or using machine learning (ML)-based surrogate approaches for fast parameter inferences [3,4].

page 15, line 46, remove “intensive”, and please rewrite “ranging from . . . to . . . ” and provide references as examples.

Corrected.

Page 15, line 46, please define machine learning as “ML”, as it is the first appearing in the main text.

Corrected.

Page 15, line 48, please write left ventricular next to “LV” as the authors first mention it in the main text.

Corrected.

Page 15, line 56, maybe also mention <http://dx.doi.org/10.1098/rsta.2019.0334>?

Corrected. Referred the paper.

Figure 1 caption, “Visualisation” instead of “Visualise”.

Corrected.

Page 16, line 56, define Ψ as the strain-energy function in the text.

Corrected.

Page 17, line 43, misplaced period, “Klotz et al.”.

Corrected.

Page 17, line 48, insert “and” before “ $\bar{\epsilon}_{\max}$ ”.

Corrected.

Page 20, Eq. (2.8), the “ $L_{1,2}$ ” notation seems confusing, maybe just “L”?

Corrected. Replaced $L_{1,2}$ with L. Here L is a distance metric function, which can be either as L_1, L_2 .

Page 20, line 41, L_1, L_2 missing subscripting, and either “a distance function” or “a metric”, but not “distance metric function”.

Corrected. Replaced “a distance metric function” with “a distance function”.

Page 20, Algorithm 1, I don't find the pseudo-algorithm adds more information than the text has already explained; maybe remove and just add a sentence before this to clarify the hyper-parameters are K and the distance function.

Corrected. Removed the pseudo-algorithm. Hyper-parameters would be clarified in Section result.

Page 21, line 9, “Chen and Guestrin [40]” (missing the second author).

Corrected.

Page 21, line 10, remove “engineering”, and spell out “GBDT”.

Corrected.

Page 21, line 12, spell out “CART”. And “CART tree” is a redundant acronym, just CART is fine (T stands for tree).

Corrected.

Page 21, line 25, again, either “a distance function” or “a metric”.

Corrected. Also replaced with “a distance function”.

Page 21, line 26, misplaced “(t)” in the target data, that should go to the first y.

Corrected. \hat{y}^i represents the target value of sampling data, and $y^{i(t)}$ represents the predicted value in current step t.

Page 23, line 10, please refer the size of hidden layer to as “number of neurons” as well for clarity.

Corrected.

Page 23, Table 1, please provide the detail of the software packages used before referring to the specific function arguments of that particular software!

Corrected

Page 23, line 53, capitalize “Python”.

Corrected.

Page 27, line 8, “. . . using the inverse approach developed. . . ”.

Corrected.

Page 27, line 19, I think it’s better to call the plots in Figure 10 as “surface plots of the objective function” instead of “distributions of objective function”, as it is not a distribution in a statistical sense, which can be confusing.

Corrected.

Page 27, line 22, missing mention of the lsqnonlin solver/ABAQUS simulation.

Corrected. Added ABAQUS simulation.

Page 27, line 27, suggest referring to Eq. 2.5 here, i.e. “. . . which may be explained by the limitation of the reduced material parameter space in Equation (2.5).”

Corrected.

Page 29, line 15, missing period, “Gao et al.”.

Corrected.

Referees: 2

Author developed three surrogate models based on three machine learning methods, the KNN, XGBoost and MLP to estimate the material parameters of left ventricle myocardium using the cavity volume and the maximum and minimum principal strains. This study is very novel and interesting addressing an important clinically relevant problem. Authors might have to discuss that in the real-world, the in vivo images are acquired with the physiological pressure, that is, the heart configuration obtained is pressurised. However, the mechanical analysis needs to be performed based on stress-free configuration, alternatively, at least based on the zero-pressure configuration. How this will affect the parameters estimation following the procedure as proposed in this study?

Thank the reviewer for the very supportive comment. We discuss this in the last second paragraph of Discussion section (page 19 line 487 – 496). It reads:

A further limitation is that the LV geometry was reconstructed from in vivo images, which is not stress free because of the non-zero blood pressure in the LV cavity. Since in this study, we do not intend to infer subject-specific material parameters using in vivo data, instead to compare the three ML models for learning generic LV dynamics in diastole and their suitability of inferring parameters by replacing expensive FE LV models. Future work should retrain the ML models with subject-specific LV models for potential clinical applications starting stress-free state, while this will require tremendous efforts for building such a large number of personalized heart models, currently not available. Still, determining the fully stress-free state of the left ventricle from in vivo data is very challenging because the heart is always pressurized. Future studies shall quantify how such a loaded geometry will affect the parameter estimation using in vivo data.

Minor comments

1. *The full description should be given when the abbreviation firstly appeared in the main text.*

Corrected

2. *It must be a typo for 'Figure 7' in the 2nd paragraph in the section (ii) strategy for output features.*

Corrected. We have redrawn this figure and updated explanations, which can be found in Figure 7 (page 12).

References

1. Gao H , Mangion K , Carrick D , et al. Estimating prognosis in patients with acute myocardial infarction using personalized computational heart models[J]. Rep, 2017, 7(1):13527.
2. Gao H , Li W G , Cai L , et al. Parameter estimation in a Holzapfel–Ogden law for healthy myocardium[J]. Journal of Engineering Mathematics, 2015, 95(1):231-248.
3. Neto, EduardodeSouza. Computational Methods for Plasticity: Theory and Applications[M]. Wiley, 2008.
4. Davies V , Umberto Noè, Lazarus A , et al. Fast Parameter Inference in a Biomechanical Model of the Left Ventricle using Statistical Emulation[J]. Applied Stats, 2019, 68.
5. Dabiri Y, Van der Velden A, Sack KL, Choy JS, Kassab GS, Guccione JM. Prediction of Left Ventricular Mechanics Using Machine Learning. Front Phys. 2019;7:117.

6. Liang L , Liu M , Martin C , et al. A deep learning approach to estimate stress distribution: a fast and accurate surrogate of finite-element analysis[J]. *Journal of The Royal Society Interface*, 2018, 15(138):20170844.
7. Di Achille P , Harouni A , Khamzin S , et al. Gaussian Process Regressions for Inverse Problems and Parameter Searches in Models of Ventricular Mechanics[J]. *Frontiers in Physiology*, 2018, 9.
8. Radau P., Lu Y., Connelly K., Paul G., Dick A.J., Wright G.A. Evaluation Framework for Algorithms Segmenting Short Axis Cardiac MRI. 2009 Jul.
9. Longobardi S , Lewalle A , Coveney S , et al. Predicting left ventricular contractile function via Gaussian process emulation in aortic-banded rats[J]. *Philosophical Transactions of The Royal Society A Mathematical Physical and Engineering Sciences*, 2020, 378(2173):20190334.
10. Noè U, Lazarus A, Gao H, et al. Gaussian process emulation to accelerate parameter estimation in a mechanical model of the left ventricle: a critical step towards clinical end-user relevance. *J R Soc Interface*. 2019;16(156):20190114.
11. Lazarus A, Gao H, Luo X, Husmeier D. 2020 Improving cardio-mechanic inference by combining in-vivo strain data with ex-vivo volume-pressure data.
12. Guyon I, Bennett K, Cawley G, Escalante HJ, Escalera S, Tin Kam Ho, Macià N, Ray B, Saeed M, Statnikov A, Viegas E. 2015 Design of the 2015 ChaLearn AutoML challenge. In 2015 International Joint Conference on Neural Networks (IJCNN) pp. 1–8.
13. Martinez-Cantin R. 2014 Bayesopt: A bayesian optimization library for nonlinear optimization, experimental design and bandits. *The Journal of Machine Learning Research* 15, 3735–3739.
14. Nogueira F. 2014– Bayesian Optimization: Open source constrained global optimization tool for Python.
15. Liu M, Liang L, Sun W. 2019 Estimation of in vivo constitutive parameters of the aortic wall using a machine learning approach. *Computer methods in applied mechanics and engineering* 347, 201–217.
16. Holzapfel GA, Ogden RW. 2009 Constitutive modelling of passive myocardium: a structurally based framework for material characterization. *Philosophical Transactions of the Royal Society A: Mathematical, Physical and Engineering Sciences* 367, 3445–3475.
17. Klotz S, Hay I, Dickstein ML, Yi GH, Wang J, Maurer MS, Kass DA, Burkhoff D. 2006 Singlebeat estimation of end-diastolic pressure-volume relationship: a novel method with potential for noninvasive application. *American Journal of Physiology-Heart and Circulatory Physiology* 291, H403–H412.
18. Mirams GR, Pathmanathan P, Gray RA, Challenor P, Clayton RH. 2016 Uncertainty and variability in computational and mathematical models of cardiac physiology. *The Journal of physiology* 594, 6833–6847.
19. Clayton RH, Aboelkassem Y, Cantwell CD, Corrado C, Delhaas T, Huberts W, Lei CL, Ni H, Panfilov AV, Roney C et al.. 2020 An audit of uncertainty in multi-scale cardiac electrophysiology models. *Philosophical Transactions of the Royal Society A* 378, 20190335
20. Lei CL, Ghosh S, Whittaker DG, Aboelkassem Y, Beattie KA, Cantwell CD, Delhaas T, Houston C, Novaes GM, Panfilov AV et al.. 2020 Considering discrepancy when calibrating a mechanistic electrophysiology model. *Philosophical Transactions of the Royal Society A* 378, 20190349.
21. Campos J, Sundnes J, Dos Santos R, Rocha B. 2020 Uncertainty quantification and sensitivity analysis of left ventricular function during the full cardiac cycle. *Philosophical Transactions of the Royal Society A* 378, 20190381.

22. Changes and classification in myocardial contractile function in the left ventricle following acute myocardial infarction.
23. Pedregosa F, Varoquaux G, Gramfort A, Michel V, Thirion B, Grisel O, Blondel M, Prettenhofer P, Weiss R, Dubourg V, Vanderplas J, Passos A, Cournapeau D, Brucher M, Perrot M, Duchesnay E. 2011 Scikit-learn: Machine Learning in Python. *Journal of Machine Learning Research* 12, 2825–2830.
24. Samet H. 1990 *The design and analysis of spatial data structures* vol. 85. Addison-Wesley Reading, MA.
25. Kingma DP, Ba J. 2017 Adam: A Method for Stochastic Optimization.